# AN UNCONSTRAINED LAYER-PEELED PERSPECTIVE ON NEURAL COLLAPSE

**Wenlong Ji**
Peking University

**Yiping Lu**
Stanford University

**Yiliang Zhang**
University of Pennsylvania

**Zhun Deng**
Harvard University

**Weijie J. Su**
University of Pennsylvania

## ABSTRACT

Neural collapse is a highly symmetric geometry of neural networks that emerges during the terminal phase of training, with profound implications on the generalization performance and robustness of the trained networks. To understand how the last-layer features and classifiers exhibit this recently discovered implicit bias, in this paper, we introduce a surrogate model called the unconstrained layer-peeled model (ULPM). We prove that gradient flow on this model converges to critical points of a minimum-norm separation problem exhibiting neural collapse in its global minimizer. Moreover, we show that the ULPM with the cross-entropy loss has a benign global landscape for its loss function, which allows us to prove that all the critical points are strict saddle points except the global minimizers that exhibit the neural collapse phenomenon. Empirically, we show that our results also hold during the training of neural networks in real-world tasks when explicit regularization or weight decay is not used.

## 1 INTRODUCTION

Deep learning has achieved state-of-the-art performance in various applications (LeCun et al., 2015), such as computer vision (Krizhevsky et al., 2012), natural language processing (Brown et al., 2020), and scientific discovery (Long et al., 2018; Zhang et al., 2018). Despite the empirical success of deep learning, how gradient descent or its variants lead deep neural networks to be biased towards solutions with good generalization performance on the test set is still a major open question. To develop a theoretical foundation for deep learning, many studies have investigated the implicit bias of gradient descent in different settings (Li et al., 2018; Amari et al., 2020; Vaswani et al., 2020; Soudry et al., 2018; Lyu & Li, 2019; Arora et al., 2019).

It is well acknowledged that well-trained end-to-end deep architectures can effectively extract features relevant to a given label. Although theoretical analysis of deep learning has been successful in recent years (Arora et al.; Goldblum et al., 2019), most of the studies that aim to analyze the properties of the final output function fail to understand the features learned by neural networks. Recently, in Papyan et al. (2020), the authors observed that the features in the same class will collapse to their mean and the mean will converge to an equiangular tight frame (ETF) during the terminal phase of training, that is, the stage after achieving zero training error. This phenomenon, namely, neural collapse (Papyan et al., 2020), provides a clear view of how the last-layer features in the neural network evolve after interpolation and enables us to understand the benefit of training after achieving zero training error to achieve better performance in terms of generalization and robustness. To theoretically analyze the neural collapse phenomenon, Fang et al. (2021) proposed the layer-peeled model (LPM) as a simple surrogate for neural networks, where the last-layer features are modeled as free optimization variables. In particular, in a balanced $K$-class classification problem using a neural network with $d$ neurons in the last hidden layer, the LPM takes the following form:

$$\min_{\boldsymbol{W},\boldsymbol{H}} \frac{1}{n}\sum_{i=1}^{n}\mathcal{L}\left(\boldsymbol{W}\boldsymbol{h}_i, \boldsymbol{y}_i\right), \quad \text{s.t.} \ \frac{1}{2}||\boldsymbol{W}||_F^2 \le C_1, \frac{1}{2}||\boldsymbol{H}||_F^2 \le C_2, \qquad (1.1)$$

where $C_1, C_2$ are positive constants. Here, $\boldsymbol{W} = [\boldsymbol{w}_1, \boldsymbol{w}_2, \cdots, \boldsymbol{w}_K]^\top \in \mathbb{R}^{K \times d}$ is the weight of the final linear classifier, $\boldsymbol{H} = [\boldsymbol{h}_1, \boldsymbol{h}_2, \cdots, \boldsymbol{h}_n] \in \mathbb{R}^{d \times n}$ is the feature of the last layer and $\boldsymbol{y}_i$ is the corresponding label. The intuition behind the LPM is that modern deep networks are often highly over-parameterized, with the capacity to learn any representations of the input data. It has been shown that an equiangular tight frame (ETF), i.e., feature with neural collapse, is the only global optimum of the LPM objective (1.1) (Fang et al., 2021; Lu & Steinerberger, 2020; Wojtowytsch & E, 2020; Zhu et al., 2021).

However, feature constraints in LPMs are not equivalent to weight decay used in practice. In this study, we directly deal with the unconstrained model and show that gradient flow can find those neural collapse solutions without the help of explicit constraints and regularization. To do this, we build a connection between the neural collapse and recent theories on max-margin implicit regularization (Lyu & Li, 2019; Wei et al., 2018), and use it to provide a convergence result to the first-order stationary point of a minimum-norm separation problem. Furthermore, we illustrate that the cross-entropy loss enjoys a benign global landscape where all the critical points are strict saddles in the tangent space, except for the only global minimizers that exhibit the neural collapse phenomenon. Finally, we verify our insights via empirical experiments.

In contrast to previous theoretical works on neural collapse, our analysis does not incorporate any explicit regularization or constraint on features. A comparison with other results can be found in Table 1 and we defer a detailed discussion to Section 5.2. The reasons we investigate the unregularized objective are summarized as follows:

1. Feature regularization or constrain is still not equivalent to weight decay used in practice. However, previous studies have justified that neural networks continue to perform well without any regularization or constraint (Zhang et al., 2021). Moreover, it is proved that SGD with exponential learning rate on unconstrained objective is equivalent to SGD with weight decay. (Li & Arora, 2019).

2. As shown in this study, neural collapse exists even under an unconstrained setting, which implies the emergence of neural collapse should be attributed to gradient descent and cross-entropy loss rather than explicit regularization.

3. Regularization or constraint feature constraint can be barriers for existing theories of neural networks (Jacot et al., 2018; Lyu & Li, 2019). By allowing features to be totally free, we hope our results can inspire further analysis to plug in a realistic neural network.

| Reference | Contribution | Feature Norm Constraint | Feature Norm Regularization | Loss Function |
|---|---|---|---|---|
| (Papyan et al., 2020) | Empirical Results | ✗ | ✗ | CE Loss |
| (Wojtowytsch & E, 2020) (Lu & Steinerberger, 2020) (Fang et al., 2021) | Global Optimum | ✔ | ✗ | CE Loss |
| (Mixon et al., 2020) (Poggio & Liao, 2020) (Han et al., 2021) | Modified Training Dynamics | ✗ | ✗ | $\ell_2$ Loss |
| (Zhu et al., 2021) **concurrent** | Landscape Analysis | ✗ | ✔ | CE Loss |
| This paper | Training Dynamics+ Landscape Analysis | ✗ | ✗ | CE Loss |

Table 1: Comparison of recent analysis for neural collapse. We provide theoretical results with the minimum modification of the training objective function. Here we use CE loss to refer to the cross-entropy loss.

## 1.1 CONTRIBUTIONS

The contributions of the present study can be summarized as follows.

- We build a relationship between the max-margin analysis (Soudry et al., 2018; Nacson et al., 2019b; Lyu & Li, 2019) and the neural collapse and provide the implicit bias analysis to the feature rather than the output function. Although both parameters and features diverge to infinity, we prove that the convergent direction is along the direction of the minimum-norm separation problem.

- Previous works (Lyu & Li, 2019; Ji et al., 2020) only prove that gradient flow on homogeneous neural networks will converge to the KKT point of the corresponding minimum-norm separation problem. However, the minimum-norm separation problem remains a highly non-convex problem and a local KKT point may not be the neural collapse solution. In this study, we perform a more detailed characterization of the convergence direction via landscape analysis.

- Previous analysis about neural collapse relies on the explicit regularization or constraint. In this study, we show that the implicit regularization effect of gradient flow is sufficient to lead to a neural collapse solution. The emergence of neural collapse should be attributed to gradient descent and loss function, rather than explicit regularization or constraint. We put detailed discussion in Section 5.2.

## 1.2 RELATED WORKS

**Implicit Bias of Gradient Descent:** To understand how gradient descent or its variants helps deep learning to find solutions with good generalization performance on the test set, a recent line of research have studied the implicit bias of gradient descent in different settings. For example, gradient descent is biased toward solutions with smaller weights under $\ell_2$ loss (Li et al., 2018; Amari et al., 2020; Vaswani et al., 2020) and will converge to large margin solution while using logistic loss (Soudry et al., 2018; Nacson et al., 2019b; Lyu & Li, 2019; Chizat & Bach, 2020; Ji et al., 2020). For linear networks, Arora et al. (2019); Razin & Cohen (2020); Gidel et al. (2019) have shown that gradient descent determines a low-rank approximation.

**Loss Landscape Analysis:** Although the practical optimization problems encountered in machine learning are often nonconvex, recent works have shown the landscape can enjoy benign properties which allow further analysis. In particular, these landscapes do not exhibit spurious local minimizers or flat saddles and can be easily optimized via gradient-based methods (Ge et al., 2015). Examples include phase retrieval (Sun et al., 2018), low-rank matrix recovery (Ge et al., 2016; 2015), dictionary learning (Sun et al., 2016; Qu et al., 2019; Laurent & Brecht, 2018) and blind deconvolution (Lau et al., 2019).

## 2 PRELIMINARIES AND PROBLEM SETUP

In this paper, $||\cdot||_F$ denotes the Frobenius norm, $||\cdot||_2$ denotes the matrix spectral norm, $||\cdot||_*$ denotes the nuclear norm, $||\cdot||$ denotes the vector $l_2$ norm and $\mathrm{tr}(\cdot)$ is the trace of matrices. We use $[K] := \{1, 2, \cdots, K\}$ to denote the set of indices up to $K$.

## 2.1 PRELIMINARIES

We consider a balanced dataset with $K$ classes $\bigcup_{k=1}^{K} \{x_{k,i}\}_{i=1}^{n}$. A standard fully connected neural network can be represented as:

$$f(x; W_{full}) = b_L + W_L \sigma(b_{L-1} + W_{L-1}\sigma(\cdots\sigma(b_1 + W_1 x)\cdots)). \tag{2.1}$$

Here $W_{full} = (W_1, W_2, \cdots, W_L)$ denote the weight matrices in each layer, $(b_1, b_2, \cdots, b_L)$ denote the bias terms, and $\sigma(\cdot)$ denotes the nonlinear activation function, for example, ReLU or sigmoid. Let $h_{k,i} = \sigma(b_{L-1} + W_{L-1}\sigma(\cdots\sigma(b_1 + W_1 x_{k,i}))) \in \mathbb{R}^d$ denote the last layer feature for data $x_{k,i}$ and $\bar{h}_k = \frac{1}{n}\sum_{i=1}^{n} h_{k,i}$ denotes the feature mean within the k-th class. To provide a formal definition of neural collapse, we first introduce the concept of a simplex equiangular tight frame (ETF):

**Definition 2.1** (Simplex ETF). A symmetric matrix $M \in \mathbb{R}^{K \times K}$ is said to be a simplex equiangular tight frame (ETF) if

$$M = \sqrt{\frac{K}{K-1}} Q(I_K - \frac{1}{K}\mathbf{1}_K\mathbf{1}_K^\top). \tag{2.2}$$

Where $\boldsymbol{Q} \in \mathbb{R}^{d \times K}$ is a matrix with orthogonal columns.

Let $\boldsymbol{W} \in \mathbb{R}^{K \times d} = \boldsymbol{W}_L = [\boldsymbol{w}_1, \boldsymbol{w}_2, \cdots, \boldsymbol{w}_K]^\top$ be the weight of the final layer classifier, the four criteria of neural collapse can be formulated precisely as:

- **(NC1) Variability collapse:** As training progresses, the within-class variation of the activation becomes negligible as these activation collapse to their class mean $\bar{\boldsymbol{h}}_k = \frac{1}{n} \sum_{i=1}^n \boldsymbol{h}_{k,i}$.

$$||\boldsymbol{h}_{k,i} - \bar{\boldsymbol{h}}_k|| = 0, \quad 1 \le k \le K.$$

- **(NC2) Convergence to Simplex ETF:** The vectors of the class-means (after centering by their global-mean) converge to having equal length, forming equal-size angles between any given pair, and being the maximally pairwise-distanced configuration constrained to the previous two properties.

$$\cos(\bar{\boldsymbol{h}}_k, \bar{\boldsymbol{h}}_j) = -\frac{1}{K-1}, \quad ||\bar{\boldsymbol{h}}_k|| = ||\bar{\boldsymbol{h}}_j||, \quad k \ne j.$$

- **(NC3) Convergence to self-duality:** The linear classifiers and class-means will converge to align with each other, up to appropriate rescaling, that is, there exist a universal constant $C > 0$ such that

$$\boldsymbol{w}_k = C\bar{\boldsymbol{h}}_k, \quad k \in [K].$$

- **(NC4) Simplification to Nearest Class-Center** For a given deepnet activation:

$$\boldsymbol{h} = \sigma\left(\boldsymbol{b}_{L-1} + \boldsymbol{W}_{L-1}\sigma\left(\cdots\sigma\left(\boldsymbol{b}_1 + \boldsymbol{W}_1\boldsymbol{x}\right)\cdots\right)\right) \in \mathbb{R}^d,$$

the network classifier converges to choose whichever class has the nearest train class-mean

$$\arg\min_k \langle \boldsymbol{w}_k, \boldsymbol{h} \rangle \to \arg\min_k \left\|\boldsymbol{h} - \bar{\boldsymbol{h}}_k\right\|,$$

In this paper, we say that a point $\boldsymbol{W} \in \mathbb{R}^{K \times d}, \boldsymbol{H} \in \mathbb{R}^{d \times nK}$ satisfies the neural collapse conditions or is a neural collapse solution if the above four criteria are all satisfied for $(\boldsymbol{W}, \boldsymbol{H})$.

## 2.2 PROBLEM SETUP

We mainly focus on the neural collapse phenomenon, which is only related to the classifiers and features in the last layer. Since the general analysis of the highly non-smooth and non-convex neural network is difficult, we peel down the last layer of the neural network and propose the following **unconstrained layer-peeled model (ULPM)** as a simplification to capture the main characters related to neural collapse during training dynamics. A similar simplification is commonly used in previous theoretical works (Lu & Steinerberger, 2020; Fang et al., 2021; Wojtowytsch & E, 2020; Zhu et al., 2021), but ours does not have any constraint or regularization on features. It should be mentioned that although Mixon et al. (2020); Han et al. (2021) also studied the unconstrained model, their analysis adopted an approximation to real dynamics and was highly dependent on the $\ell_2$ loss function, which is rarely used in classification tasks. Compared with their works, we directly deal with the real training dynamics and cover the most popular cross-entropy loss in classification tasks.

Let $\boldsymbol{W} = [\boldsymbol{w}_1, \boldsymbol{w}_2, \cdots, \boldsymbol{w}_K]^\top \in \mathbb{R}^{K \times d}$ and $\boldsymbol{H} = [\boldsymbol{h}_{1,1}, \cdots, \boldsymbol{h}_{1,n}, \boldsymbol{h}_{2,1}, \cdots, \boldsymbol{h}_{K,n}] \in \mathbb{R}^{d \times Kn}$ be the matrices of classifiers and features in the last layer, where $K$ is the number of classes and $n$ is the number of data points in each class. The ULPM is defined as follows:

$$\min_{\boldsymbol{W}, \boldsymbol{H}} \mathcal{L}(\boldsymbol{W}, \boldsymbol{H}) = \min_{\boldsymbol{W}, \boldsymbol{H}} -\sum_{k=1}^K \sum_{i=1}^n \log\left(\frac{\exp(\boldsymbol{w}_k^\top \boldsymbol{h}_{k,i})}{\sum_{j=1}^K \exp(\boldsymbol{w}_j^\top \boldsymbol{h}_{k,i})}\right). \quad (2.3)$$

Here, we do not have any constraint or regularization on features, which corresponds to the absence of weight decay in deep learning training. The objective function (2.3) is generally non-convex on $(\boldsymbol{W}, \boldsymbol{H})$ and we aim to study the landscape of the objective function (2.3). Furthermore, we consider the gradient flow of the objective function:

$$\frac{d\boldsymbol{W}(t)}{dt} = -\frac{\partial\mathcal{L}(\boldsymbol{W}(t), \boldsymbol{H}(t))}{\partial\boldsymbol{W}}, \frac{d\boldsymbol{H}}{dt} = -\frac{\partial\mathcal{L}(\boldsymbol{W}(t), \boldsymbol{H}(t))}{\partial\boldsymbol{H}}.$$

# 3    MAIN RESULTS

In this section, we present our main results regarding the training dynamics and landscape analysis of (2.3). This section is organized as follows: First, in Section 3.1, we show the relationship between the margin and neural collapse in our surrogate model. Inspired by this relationship, we propose a minimum-norm separation problem (3.1) and show the connection between the convergence direction of the gradient flow and the KKT point of (3.1). In addition, we explicitly solve the global optimum of (3.1) and show that it must satisfy the neural collapse conditions. However, owing to the non-convexity, we find an Example 3.1 in Section 3.2, which shows that there exist some bad KKT points such that a simple gradient flow will get stuck in them and does not converge to the neural collapse solution, which is proved to be optimal in Theorem 3.3. Then, we present our second–order analysis result in Theorem 3.4 to show that those bad points will exhibit decreasing directions in the tangent space thus gradient descent and its variants can avoid those bad directions and only converge to the neural collapse solutions (Lee et al., 2016; Ge et al., 2015).

## 3.1    CONVERGENCE TO THE FIRST–ORDER STATIONARY POINT

Heuristically speaking, the simplex ETF (Definition 2.1) gives a set of vectors with the maximum average angle between them. As a result, neural collapse implies that the neural networks tend to maximize the angles between each class and the corresponding classifiers. At a high level, such behavior is quite similar to margin maximization which is known to be an implicit regularization effect of gradient descent and has been extensively studied in Soudry et al. (2018); Nacson et al. (2019b); Lyu & Li (2019); Ji et al. (2020). First, we illustrate the connection between the margin and neural collapse. Recall that the margin of a single data point $\boldsymbol{x}_{k,i}$ and the associated feature $\boldsymbol{h}_{k,i}$ is $q_{k,i}(\boldsymbol{W}, \boldsymbol{H}) := \boldsymbol{w}_k^\top \boldsymbol{h}_{k,i} - \max_{j \neq k} \boldsymbol{w}_j^\top \boldsymbol{h}_{k,i}$. Then, the margin of the entire dataset can be defined as

$$q_{\min}(\boldsymbol{W}, \boldsymbol{H}) := \min_{k \in [K], i \in [n]} q_{k,i}(\boldsymbol{W}, \boldsymbol{H}).$$

The following theorem demonstrates that neural collapse yields the maximum margin solution of our ULPM model:

**Theorem 3.1** (Neural collapse as max-margin solution)**.** For the ULPM model (2.3), the margin of the entire dataset always satisfies

$$q_{\min}(\boldsymbol{W}, \boldsymbol{H}) \leq \frac{\|\boldsymbol{W}\|_F^2 + \|\boldsymbol{H}\|_F^2}{2(K-1)\sqrt{n}}$$

and the equality holds if and only if $(\boldsymbol{W}, \boldsymbol{H})$ satisfies the neural collapse conditions with $\|W\|_F = \|H\|_F$.

Based on this finding, we present an analysis of the convergence of the gradient flow on the ULPM (2.3). Following Lyu & Li (2019), we link the gradient flow on cross-entropy loss to a minimum-norm separation problem.

**Theorem 3.2.** For problem (2.3), let $(\boldsymbol{W}(t), \boldsymbol{H}(t))$ be the path of gradient flow at time $t$, if there exist a time $t_0$ such that $\mathcal{L}(\boldsymbol{W}(t_0), \boldsymbol{H}(t_0)) < \log 2$, then any limit point of

$$\{(\hat{\boldsymbol{H}}(t), \hat{\boldsymbol{W}}(t)) := (\frac{\boldsymbol{H}(t)}{\sqrt{\|\boldsymbol{W}(t)\|_F^2 + \|\boldsymbol{H}(t)\|_F^2}}, \frac{\boldsymbol{W}(t)}{\sqrt{\|\boldsymbol{W}(t)\|_F^2 + \|\boldsymbol{H}(t)\|_F^2}})\}$$

is along the direction (i.e., a constant multiple) of a Karush-Kuhn-Tucker (KKT) point of the following minimum-norm separation problem:

$$\min_{W,H} \frac{1}{2}\|\boldsymbol{W}\|_F^2 + \frac{1}{2}\|\boldsymbol{H}\|_F^2 \tag{3.1}$$
$$s.t. \boldsymbol{w}_k^\top \boldsymbol{h}_{k,i} - \boldsymbol{w}_j^\top \boldsymbol{h}_{k,i} \geq 1, \quad k \neq j \in [K], i \in [n].$$

*Remark* 3.1. Here we write (3.1) as a constraint problem, but the constraint is introduced by the implicit regularization effect of gradient flow on our ULPM objective (2.3). Since the training dynamics will diverge to infinity, we hope to justify that the diverge direction is highly related to neural collapse and an appropriate normalization is needed, which is why it appears to be a

constraint optimization form. Our goal is to justify that the neural collapse phenomenon is caused by the properties of the loss function and training dynamics rather than an explicit regularization or constraint, which seems to be necessary for previous studies (Fang et al., 2021; Lu & Steinerberger, 2020; Wojtowytsch & E, 2020).

*Remark* 3.2. In Theorem 3.2, we assume that there exists a time $t_0$ such that $\mathcal{L}(\boldsymbol{W}(t_0), \boldsymbol{H}(t_0)) < \log 2$. Note that the loss function can be rewritten as:

$$\mathcal{L}(\boldsymbol{W}, \boldsymbol{H}) = \sum_{k=1}^{K} \sum_{i=1}^{n} \log(1 + \sum_{j \neq k} \exp(\boldsymbol{w}_j \boldsymbol{h}_{k,i} - \boldsymbol{w}_k \boldsymbol{h}_{k,i})) \tag{3.2}$$

the requirement $\mathcal{L}(\boldsymbol{W}, \boldsymbol{H}) < \log 2$ implies that $\boldsymbol{w}_k \boldsymbol{h}_{k,i} - \boldsymbol{w}_j \boldsymbol{h}_{k,i} \geq 0$ for any $k, j \in [K], i \in [n]$, which is equivalent to $q_{min}(\boldsymbol{W}, \boldsymbol{H}) > 0$; that is, every feature is separated perfectly by the classifier. This assumption is common in the study of implicit bias in the nonlinear setting (Nacson et al., 2019a; Lyu & Li, 2019; Ji et al., 2020) and its validity can be justified by the fact that neural collapse is found only in the terminal phase of training in the deep neural network, where the training accuracy has achieved 100%. It is also an interesting direction to remove this assumption and study the early-stage dynamics of training, which is beyond the scope of this study and we leave it to future exploration.

Theorem 3.2 indicates that the convergent direction of the gradient flow is restricted to the max-margin directions, which usually have good robustness and generalization performance. In general, the KKT conditions are not sufficient to obtain global optimality because the minimum-norm separation problem (3.1) is non-convex. On the other hand, we can precisely characterize its global optimum in the ULPM case based on Theorem 3.1:

**Corollary 3.1.** Every global optimum of the minimum-norm separation problem (3.1) is also a KKT point that satisfies the neural collapse conditions.

With Theorem 3.2 bridging dynamics with KKT points of (3.1) and Corollary 3.1 associating global optimum of (3.1) with neural collapse, the remaining work is to close the gap between the KKT point and the global optimum.

## 3.2 Second–Order Landscape Analysis

In the convex optimization problem, the KKT conditions are usually equivalent to global optimality. Unfortunately, owing to the non-convex nature of the objective (2.3), the KKT points can also be saddle points or local optimum other than the global optimum. In this section, we aim to show that this non-convex optimization problem is actually not scary via landscape analysis. To be more specific, we prove that except for the global optimum given by neural collapse, all the other KKT points are actually saddle points that can be avoided by gradient flow.

In contrast to previous landscape analysis of non-convex problems, where people aim to show that the objective has a negative directional curvature around any stationary point (Sun et al., 2015; Zhang et al., 2020), our analysis is slightly different. Note that since the model is unconstrained, once features can be perfectly separated, the ULPM objective (2.3) will always decrease along the direction of the current point and the optimum is attained only in infinity. Although growing along all of those perfectly separating directions can let the loss function decrease to 0, the speed of decrease is quite different and there exists an optimal direction with the fastest decreasing speed. As shown in Section 3.1, first-order analysis of training dynamics fails to distinguish such an optimal direction from KKT points, and we need second-order analysis to help us fully characterize the realistic training dynamics. First, we provide an example to illustrate the motivation and necessity of second-order landscape analysis.

**Example 3.1** (A Motivating Example). Consider the case where $K = 4, n = 1$, let $(\boldsymbol{W}, \boldsymbol{H})$ be the following point:

$$\boldsymbol{W} = \boldsymbol{H} = \begin{bmatrix} 1 & -1 & 0 & 0 \\ -1 & 1 & 0 & 0 \\ 0 & 0 & 1 & -1 \\ 0 & 0 & -1 & 1 \end{bmatrix}.$$

One can easily verify that $(\boldsymbol{W}, \boldsymbol{H})$ enables our model to classify all features perfectly. Furthermore, we can show that it is along the direction of a KKT point of the minimum-norm separation problem

(3.1) by constructing the Lagrangian multiplier $\Lambda = (\lambda_{ij})_{i,j=1}^K$ as follows:

$$\Lambda = \begin{bmatrix} 0 & 0 & \frac{1}{2} & \frac{1}{2} \\ 0 & 0 & \frac{1}{2} & \frac{1}{2} \\ \frac{1}{2} & \frac{1}{2} & 0 & 0 \\ \frac{1}{2} & \frac{1}{2} & 0 & 0 \end{bmatrix}.$$

To see this, simply write down the corresponding Lagrangian (note that we aim to justify $(\boldsymbol{W}, \boldsymbol{H})$ is along the direction of a KKT point of (3.1), to make it a true KKT point , one needs to multiple $1/\sqrt{2}$ on $\boldsymbol{W}, \boldsymbol{H}$):

$$\mathcal{L}(\boldsymbol{W}, \boldsymbol{H}, \Lambda) = \frac{1}{4}\|\boldsymbol{W}\|_F^2 + \frac{1}{4}\|\boldsymbol{H}\|_F^2 - \sum_{i=1}^4 \sum_{j \neq i} \lambda_{i,j}(\frac{1}{2}\boldsymbol{w}_i\boldsymbol{h}_i - \frac{1}{2}\boldsymbol{w}_j\boldsymbol{h}_i - 1).$$

Simply take derivatives for $\boldsymbol{W}, \boldsymbol{H}$ and $\Lambda$ we find that it satisfies the KKT conditions. However, the gradient of $(\boldsymbol{W}, \boldsymbol{H})$ is:

$$\nabla_{\boldsymbol{W}}\mathcal{L}(\boldsymbol{W}, \boldsymbol{H}) = \nabla_{\boldsymbol{H}}\mathcal{L}(\boldsymbol{W}, \boldsymbol{H}) = -\frac{2 + 2e^{-2}}{2 + 2e^{-2} + 2e^2} \begin{bmatrix} 1 & -1 & 0 & 0 \\ -1 & 1 & 0 & 0 \\ 0 & 0 & 1 & -1 \\ 0 & 0 & -1 & 1 \end{bmatrix}.$$

We can see that the directions of the gradient and the parameter align with each other (i.e., $\boldsymbol{W}$ is parallel to $\nabla_{\boldsymbol{W}}\mathcal{L}(\boldsymbol{W}, \boldsymbol{H})$, and $\boldsymbol{H}$ is parallel to $\nabla_{\boldsymbol{H}}\mathcal{L}(\boldsymbol{W}, \boldsymbol{H})$), which implies that simple gradient descent may get stuck in this direction and only grow the parameter norm. However, if we construct:

$$\boldsymbol{W}' = \boldsymbol{H}'^\top = \sqrt{\frac{1}{1 + 2\alpha^2}} \begin{bmatrix} 1 + \alpha & -1 + \alpha & \alpha & \alpha \\ -1 + \alpha & 1 + \alpha & \alpha & \alpha \\ -\alpha & -\alpha & 1 - \alpha & -1 - \alpha \\ -\alpha & -\alpha & -1 - \alpha & 1 - \alpha \end{bmatrix},$$

By simple calculation, we find that $f(\alpha) := \mathcal{L}(\boldsymbol{W}', \boldsymbol{H}')$ satisfies $f'(\alpha) = 0, f''(\alpha) < 0$. Since $\|\boldsymbol{W}'\|_F = \|\boldsymbol{W}\|_F, \|\boldsymbol{H}'\|_F = \|\boldsymbol{H}\|_F$ and $\|\boldsymbol{W}' - \boldsymbol{W}\|_F^2 + \|\boldsymbol{H}' - \boldsymbol{H}\|_F^2 \to 0$ as $\alpha \to 0$, this result implies that for any $\epsilon > 0$, we can choose appropriate $\alpha$ such that:

$$\|\boldsymbol{W}'\|_F^2 = \|\boldsymbol{W}\|_F^2, \|\boldsymbol{H}'\|_F^2 = \|\boldsymbol{H}\|_F^2,$$
$$\|\boldsymbol{W}' - \boldsymbol{W}\|_F^2 + \|\boldsymbol{H}' - \boldsymbol{H}\|_F^2 < \epsilon, \mathcal{L}(\boldsymbol{W}', \boldsymbol{H}') < \mathcal{L}(\boldsymbol{W}, \boldsymbol{H}).$$

In Example 3.1, it is shown that there are some KKT points of the minimum-norm separation problem (3.1) that are not globally optimal, but there exist some better points close to it; thus, the gradient-based method can easily avoid them (see Lee et al. (2016); Ge et al. (2015) for a detailed discussion). In the following theorem, we will show that the best directions are neural collapse solutions in the sense that the loss function is the lowest among all the growing directions.

**Theorem 3.3.** The optimal value of the loss function (2.3) on a sphere is obtained (i.e., $\mathcal{L}(\boldsymbol{W}, \boldsymbol{H}) \leq \mathcal{L}(\boldsymbol{W}', \boldsymbol{H}')$ for any $\|\boldsymbol{W}'\|_F^2 + \|\boldsymbol{H}'\|_F^2 = \|\boldsymbol{W}\|_F^2 + \|\boldsymbol{H}\|_F^2$) if only if $(\boldsymbol{W}, \boldsymbol{H})$ satisfies neural collapse conditions and $\|\boldsymbol{W}\|_F = \|\boldsymbol{H}\|_F$.

*Remark* 3.3. Note that the second condition is necessary because neural collapse conditions do not specify the norm ratio of $\boldsymbol{W}$ and $\boldsymbol{H}$. That is, if $(\boldsymbol{W}, \boldsymbol{H})$ satisfies the neural collapse conditions, then for any $\alpha, \beta \in \mathbb{R}, (\alpha\boldsymbol{W}, \beta\boldsymbol{H})$ will also satisfy them, but only some certain $\alpha, \beta$ are optimal.

Now, we turn to those points that are not globally optimal. To formalize our discussion in the motivating example 3.1, we first introduce the tangent space:

**Definition 3.1** (tangent space). The tangent space of $(\boldsymbol{W}, \boldsymbol{H})$ is defined as a set of directions that are orthogonal to $(\boldsymbol{W}, \boldsymbol{H})$ :

$$\mathcal{T}(\boldsymbol{W}, \boldsymbol{H}) = \{\Delta\boldsymbol{W} \in \mathbb{R}^{K \times d}, \Delta\boldsymbol{H} \in \mathbb{R}^{d \times nK}) : \text{tr}(\boldsymbol{W}^\top \Delta\boldsymbol{W}) + \text{tr}(\boldsymbol{H}^\top \Delta\boldsymbol{H}) = 0\}$$

Our next result justifies our observation in Example 3.1 that for those non-optimal points, there exists a direction in the tangent space such that moving along this direction will lead to a lower objective value.

**Theorem 3.4.** If $(\boldsymbol{W}, \boldsymbol{H})$ is not the optimal solution in Theorem 3.3 (i.e., $(\boldsymbol{W}, \boldsymbol{H})$ is not a neural collapse solution or it is a neural collapse solution but $\|\boldsymbol{W}\|_F \neq \|\boldsymbol{H}\|_F$), then there exists a direction $(\Delta\boldsymbol{W}, \Delta\boldsymbol{H}) \in \mathcal{T}(\boldsymbol{W}, \boldsymbol{H})$ and constant $M > 0$ such that for any $0 < \delta < M$,

$$\mathcal{L}(\boldsymbol{W} + \delta\Delta\boldsymbol{W}, \boldsymbol{H} + \delta\Delta\boldsymbol{H}) < \mathcal{L}(\boldsymbol{W}, \boldsymbol{H}). \tag{3.3}$$

Further more, it implies that for any $\epsilon > 0$ there exists a point $(\boldsymbol{W}', \boldsymbol{H}')$ such that:

$$\begin{aligned}
\|\boldsymbol{W}'\|_F^2 + \|\boldsymbol{H}'\|_F^2 &= \|\boldsymbol{W}\|_F^2 + \|\boldsymbol{H}\|_F^2, \\
\|\boldsymbol{W}' - \boldsymbol{W}\|_F^2 + \|\boldsymbol{H}' - \boldsymbol{H}\|_F^2 &< \epsilon, \mathcal{L}(\boldsymbol{W}', \boldsymbol{H}') < \mathcal{L}(\boldsymbol{W}, \boldsymbol{H}).
\end{aligned} \tag{3.4}$$

*Remark* 3.4. The result in (3.3) gives us a decreasing direction orthogonal to the direction of $(\boldsymbol{W}, \boldsymbol{H})$. As shown in Example 3.1, the gradient on these non-optimal points might be parallel to $(\boldsymbol{W}, \boldsymbol{H})$; thus, the first-order analysis fails to explain the prevalence of neural collapse. Here the decreasing direction is obtained by analyzing the Riemannian Hessian matrix and finding its eigenvector corresponding to a negative eigenvalue, which further indicates that these points are first-order saddle points in the tangent space. That is why we name it second-order landscape analysis. A formal statement and proof are presented in the appendix C. Previous works have shown that for a large family of gradient-based methods, they can avoid saddle points and only converge to minimizers (Lee et al., 2016; Ge et al., 2015; Panageas et al., 2019), thus our landscape analysis indicates that the gradient flow dynamics only find neural collapse directions.

## 4 EMPIRICAL RESULTS

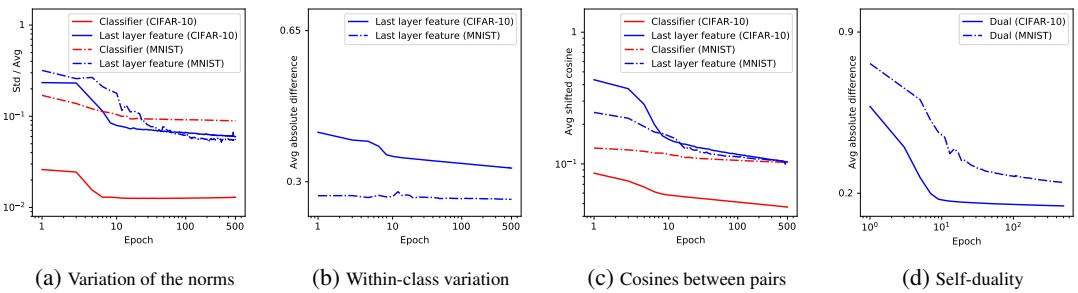

(a) Variation of the norms     (b) Within-class variation     (c) Cosines between pairs     (d) Self-duality

Figure 1: Experiments on real datasets without weight decay. We trained a ResNet18 on both MNIST and CIFAR10 datasets. The $x$-axis in the figures are set to have $\log(\log(t))$ scales and the $y$-axis in the figures are set to have $\log$ scales.

To evaluate our theory, we trained the ResNet18 (He et al., 2016) on both MNIST (LeCun et al., 1998) and CIFAR-10 (Krizhevsky et al., 2009) datasets without weight decay, and tracked how the last layer features and classifiers converge to neural collapse solutions. The results are plotted in Figure 1. Here we reported the following metrics to measure the level of neural collapse:

1. The variation of both feature norm (i.e., $\text{Std}(\|\bar{\boldsymbol{h}}_k - \bar{\boldsymbol{h}}\|)/\text{Avg}(\|\bar{\boldsymbol{h}}_k - \bar{\boldsymbol{h}}\|)$) and classifier norm in the last layer (i.e., $\text{Std}(\|\bar{\boldsymbol{w}}_k\|)/\text{Avg}(\|\bar{\boldsymbol{w}}_k\|)$).

2. Within-class variation for last layer features (i.e., $\text{Avg}(\|\boldsymbol{h}_{k,i} - \boldsymbol{h}_k\|)/\text{Avg}(\|\boldsymbol{h}_{k,i} - \bar{\boldsymbol{h}}\|)$).

3. Average cosine between last layer features (i.e., $\text{Avg}(|\cos(\bar{\boldsymbol{h}}_k, \bar{\boldsymbol{h}}_{k'}) + 1/(K-1)|)$) and that of last layer classifiers (i.e., $\text{Avg}(|\cos(\bar{\boldsymbol{w}}_k, \bar{\boldsymbol{w}}_{k'}) + 1/(K-1)|)$).

4. Self-duality distance between features and classifiers corresponding to the same class in the last layer. (i.e., $\text{Avg}(|(\bar{\boldsymbol{h}}_k - \bar{\boldsymbol{h}})/\|\bar{\boldsymbol{h}}_k - \bar{\boldsymbol{h}}\| - \bar{\boldsymbol{w}}_k/\|\bar{\boldsymbol{w}}_k\||)$)

Here a smaller value of each metric indicates a closer state to a neural collapse solution. Specifically, the within-class variation measures (NC1), the variation of norms measures and cosine between pairs measure (NC2), the self-duality measures (NC3), and (NC4) has been shown to be a corollary of (NC1)-(NC3) (Papyan et al., 2020). More experiment results under various settings and training details can be found in Appendix D. As shown in the Figure 1, all of these metric decreases along the training epochs. These results reveal strong evidence for the emergence of neural collapse in the unconstrained setting and provide sound support for our theory.

# 5 CONCLUSION AND DISCUSSION

## 5.1 CONCLUSION

To understand the implicit bias of neural features from gradient descent training, we built a connection between max-margin implicit bias and the neural collapse phenomenon and studied the ULPM in this study. We proved that the gradient flow of the ULPM converges to the KKT point of a minimum-norm separation problem, where the global optimum satisfies the neural collapse conditions. Although the ULPM is non-convex, we show that ULPM has a benign landscape where all the stationary points are strict saddle points in the tangent space, except for the global neural collapse solution. Our study helps to demystify the neural collapse phenomenon, which sheds light on the generalization and robustness during the terminal phase of training deep networks in classification problems.

## 5.2 RELATIONSHIP WITH OTHER RESULTS ON NEURAL COLLAPSE

Theoretical analysis of neural collapse was first provided by Lu & Steinerberger (2020); Wojtowytsch & E (2020); Fang et al. (2021), who showed that the neural collapse solution is the only global minimum of the simplified non-convex objective function. In particular, Wojtowytsch & E (2020); Lu & Steinerberger (2020) studied a continuous integral form of the loss function and showed that the features learned should have a uniform distribution on the sphere. A more realistic discrete setting was studied in Fang et al. (2021), where the constraint is on the entire feature matrix rather than individual features. Our result utilizes the implicit bias of the cross-entropy loss function to remove the feature norm constraint, which is not practical in real applications.

Although the global optimum can be fully characterized by neural collapse conditions, the ULPM objective is still highly non-convex. Regarding optimization, Mixon et al. (2020); Poggio & Liao (2020); Han et al. (2021) analyzed the unconstrained feature model with $\ell_2$ loss and established convergence results for collapsed features for gradient descent. However, they fail to generalize to the more practical cross-entropy loss functions used in classification tasks. The analysis relies highly on the $\ell_2$ loss to obtain a closed-form gradient flow and still requires some additional approximation to guarantee global convergence.

The most relevant study is a **concurrent** work (Zhu et al., 2021), which provides a landscape analysis of the regularized unconstrained feature model. Zhu et al. (2021) turns the feature norm constraint in Fang et al. (2021) into feature norm regularization and still preserves the neural collapse global optimum. At the same time, it shows that the modified regularized objective shares a benign landscape, where all the critical points are strict saddles except for the global one. Although our study and Zhu et al. (2021) discover similar landscape results, we believe our characterization remains closer to the real algorithms since we do not introduce any constraints or regularization on the feature norm following the conventional setting in realist training. The regularization of features introduced in Zhu et al. (2021) is still different from weight decay regularization (Krogh & Hertz, 1992). However, weight decay on homogeneous neural networks is equivalent to gradient descent with scaling step size on an unregularized objective (Li & Arora, 2019; Zhang et al., 2018). Moreover, neural networks are found to perform well in the absence of weight decay, which highlights the importance of implicit regularization (Zhang et al., 2021). As a result, instead of explicit feature norm constraint/regularization, in this study, we consider implicit regularization brought by the gradient flow on cross-entropy. We show that implicit regularization is sufficient to lead the dynamics to converge to the neural collapse solution without the explicit regularization.

## ACKNOWLEDGMENTS

We are grateful to Qing Qu and X.Y. Han for helpful discussions and feedback on an early version of the manuscript. Wenlong Ji is partially supported by the elite undergraduate training program of the School of Mathematical Sciences at Peking University. Yiping Lu is supported by the Stanford Interdisciplinary Graduate Fellowship (SIGF).

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

## A    ELEMENTS OF OPTIMIZATION

In this section, we introduce some basic definitions and theory about optimization. In the following discussion, we consider a standard form inequality constrained optimization problem:

$$\min_{x \in \mathbb{R}^d} f(x)$$
$$\text{s.t. } g_i(x) \leq 0, \quad i \in [n]. \tag{A.1}$$

In addition, we assume all of those functions $f$ and $g_i$ are twice differentiable. A point $x \in \mathbb{R}^d$ is said to be feasible if and only if it satisfies all of the constraints in (A.1), i.e., $g_i(x) \leq 0, \quad i \in [n]$. And the Lagrangian of problem (A.1) is defined as following:

$$L(x, \lambda) = f(x) + \sum_{i=1}^{n} \lambda_i g_i(x).$$

### A.1    KARUSH KUHN TUCKER CONDITIONS

Now let's first introduce the definition of Karush Kuhn Tucker (KKT) point and approximate KKT point. Here we follows the definition of $(\epsilon, \delta)$-KKT point as in Lyu & Li (2019).

**Definition A.1** (Definition of KKT point). A feasible point is said to be KKT point of problem (A.1) if there exist $\lambda_1, \lambda_2, \cdots, \lambda_n \geq 0$ such that the following Karush Kuhn Tucker (KKT) conditions hold:

1. $\nabla f(x) + \sum_{i=1}^{n} \lambda_i \nabla g_i(x) = 0$

2. $\lambda_i g_i(x) = 0, \quad i \in [n]$

**Definition A.2** (Definition of $(\epsilon, \delta)$-KKT point). For any $\epsilon, \delta > 0$, a feasible point of (A.1) is said to be $(\epsilon, \delta)$-KKT point of problem (A.1) if there exist $\lambda_1, \lambda_2, \cdots, \lambda_n \geq 0$ such that:

1. $\|\nabla f(x) + \sum_{i=1}^{n} \lambda_i \nabla g_i(x)\| \leq \epsilon$

2. $\lambda_i g_i(x) \geq -\delta, \quad i \in [n]$

Generally speaking, KKT conditions might not be necessary for global optimality. We need some additional regular conditions to make it necessary. For example, as shown in Dutta et al. (2013) we can require the problem to satisfy the following Mangasarian-Fromovitz constraint qualification (MFCQ):

**Definition A.3** (Mangasarian-Fromovitz constraint qualification (MFCQ) ). For a feasible point $x$ of (A.1), problem (A.1) is said to satisfy (MFCQ) at $x$ if there exist a vector $v \in \mathbb{R}^d$ such that:

$$\langle \nabla_x g_i(x), v \rangle > 0, \quad i \in [n]. \tag{A.2}$$

Moreover, when MFCQ holds we can build a connection between approximate KKT point and KKT point, see detailed proof in Dutta et al. (2013):

**Theorem A.1** (Relationship between Approximate KKT point and KKT point). Let $\{x_k \in \mathbb{R}^d : k \in \mathbb{N}\}$ be a sequence of feasible points of (A.1) , $\{\epsilon_k > 0 : k \in \mathbb{N}\}$ and $\{\delta_k > 0 : k \in \mathbb{N}\}$ be two sequences of real numbers such that $x_k$ is an $(\epsilon_k, \delta_k)$-KKT point for every $k$, and $\epsilon_k \to 0, \delta_k \to 0$. If $x_k \to x$ as $k \to +\infty$. If MFCQ (A.3) holds at $x$, then $x$ is a KKT point of $(P)$.

## B    OMITTED PROOFS FROM SECTION 3.1

Recall our ULPM problem:

$$\min_{\boldsymbol{W}, \boldsymbol{H}} \mathcal{L}(\boldsymbol{W}, \boldsymbol{H}) = -\sum_{k=1}^{K} \sum_{i=1}^{n} \log \left( \frac{\exp(\boldsymbol{w}_k^\top \boldsymbol{h}_{k,i})}{\sum_{j=1}^{K} \exp(\boldsymbol{w}_j^\top \boldsymbol{h}_{k,i})} \right).$$

Let's review some basic notations defined in the main body. Let $s_{k,i,j} = \boldsymbol{w}_k^\top \boldsymbol{h}_{k,i} - \boldsymbol{w}_j^\top \boldsymbol{h}_{k,i}, \forall k \in [K], i \in [n], j \in [K]$, the margin of a single feature $\boldsymbol{h}_{k,i}$ is defined to be $q_{k,i}(\boldsymbol{W}, \boldsymbol{H}) := \min_{j \neq k} s_{k,i,j} = \boldsymbol{w}_k^\top \boldsymbol{h}_{k,i} - \max_{j \neq k} \boldsymbol{w}_j^\top \boldsymbol{h}_{k,i}$. We define the margin of entire dataset as $q_{min} = q_{\min}(\boldsymbol{W}, \boldsymbol{H}) = \min_{k \in [1,K], i \in [1,n]} q_{k,i}(\boldsymbol{W}, \boldsymbol{H})$. We first prove Theorem 3.1 in the mainbody.

*Proof of Theorem 3.1.* First we can find that the margin will not change if we minus a vector $a$ for all $w_j$, so if we denote the mean of classifier $\tilde{\boldsymbol{w}}_i = \boldsymbol{w}_i - \frac{1}{K} \sum_{i=1}^K \boldsymbol{w}_i$ and then we have $\boldsymbol{w}_k^\top \boldsymbol{h}_{k,i} - \max_{j \neq k} \boldsymbol{w}_j^\top \boldsymbol{h}_{k,i} = \tilde{\boldsymbol{w}}_k^\top \boldsymbol{h}_{k,i} - \max_{j \neq k} \tilde{\boldsymbol{w}}_j^\top \boldsymbol{h}_{k,i} \geq q_{min}(\boldsymbol{W}, \boldsymbol{H})$, that is:

$$\tilde{\boldsymbol{w}}_k^\top \boldsymbol{h}_{k,i} - \tilde{\boldsymbol{w}}_j^\top \boldsymbol{h}_{k,i} \geq q_{min}(\boldsymbol{W}, \boldsymbol{H}), \forall j \neq k \in [K], i \in [n]. \tag{B.1}$$

Note that $\sum_{j=1}^K \tilde{\boldsymbol{w}}_j^\top \boldsymbol{h}_{k,i} = 0$ then sum this inequality over $j$ we have:

$$(K-1)\tilde{\boldsymbol{w}}_k^\top \boldsymbol{h}_{k,i} - \sum_{j \neq k} \tilde{\boldsymbol{w}}_j^\top \boldsymbol{h}_{k,i} = K\tilde{\boldsymbol{w}}_k^\top \boldsymbol{h}_{k,i} \geq (K-1)q_{min}(\boldsymbol{W}, \boldsymbol{H}), \forall k \in [K], i \in [n].$$

By Cauchy inequality, we have:

$$\frac{1}{2}\left(\frac{1}{\sqrt{n}}||\tilde{\boldsymbol{w}}_k||_2^2 + \sqrt{n}||\boldsymbol{h}_{k,i}||_2^2\right) \geq \tilde{\boldsymbol{w}}_k^\top \boldsymbol{h}_{k,i} \geq \frac{K-1}{K}q_{min}(\boldsymbol{W}, \boldsymbol{H}). \tag{B.2}$$

Sum (B.2) over k and i we have:

$$\frac{1}{2}\sqrt{n}(||\tilde{\boldsymbol{W}}||_F^2 + ||\boldsymbol{H}||_F^2) \geq n(K-1)q_{min}(\boldsymbol{W}, \boldsymbol{H}). \tag{B.3}$$

On the other hand, we know that:

$$||\tilde{\boldsymbol{W}}||_F^2 = \sum_{i=1}^K ||\boldsymbol{w}_i - \frac{1}{K}\sum_{i=1}^K \boldsymbol{w}_i||_2^2 \leq \sum_{i=1}^K ||\boldsymbol{w}_i||_2^2 = ||\boldsymbol{W}||_F^2.$$

Then we can conclude that:

$$q_{\min}(\boldsymbol{W}, \boldsymbol{H}) \leq \frac{||\boldsymbol{W}||_F^2 + ||\boldsymbol{H}||_F^2}{2(K-1)\sqrt{n}}$$

as desired. When the equality holds, first we have $||\tilde{\boldsymbol{W}}||_F^2 = ||\boldsymbol{W}||_F^2$ which is equivalent to $\frac{1}{K}\sum_{i=1}^K w_i = 0, \tilde{w}_i = w_i$. Take it back into (B.3), then we must have all of the equality holds in (B.2) and (B.1), which give us:

$$\boldsymbol{w}_k = \sqrt{n}\boldsymbol{h}_{k,i}, ||\boldsymbol{w}_k||_2^2 = n||\boldsymbol{h}_{k,i}||_2^2 = \frac{||\boldsymbol{W}||_F^2 + ||\boldsymbol{H}||_F^2}{2K}.$$

Take this into (B.1) we have:

$$\boldsymbol{h}_{k,i} = \boldsymbol{h}_{k,i'}, \boldsymbol{h}_{k,i}^\top \boldsymbol{h}_{j,i'} = \boldsymbol{w}_k^\top \boldsymbol{w}_j = -\frac{||\boldsymbol{W}||_F^2 + ||\boldsymbol{H}||_F^2}{2K(K-1)\sqrt{n}},$$

which implies neural collapse conditions. $\qquad\square$

Now let's turn to the training dynamics and prove Theorem 3.2. Before starting the proof, we need to introduce some additional notations. Since the $\boldsymbol{W}$ and $\boldsymbol{H}$ are all optimization variables here, we denote $\theta = vec(\boldsymbol{W}, \boldsymbol{H})$ as the whole parameter for simplicity and all of previous function can be defined on $\theta$ by matching the corresponding parameters. Denote $\rho = ||\theta||$ as the norm of $\theta$ and $\tilde{\gamma} = \frac{-\log(e^{\mathcal{L}(\theta)} - 1)}{\rho^2}$. Now we can state our first lemma to show how training dynamics of gradient flow on ULPM objective (B) is related to a KKT point of (3.1).

**Lemma B.1.** If there exist a time $t_0$ such that $\mathcal{L}(\theta(t_0)) < \log 2$, then for any $t > t_0$ $\tilde{\theta} := \theta/q_{\min}(\theta)^{1/2}$ is an $(\epsilon, \delta)$ - approximate KKT point of the following minimum-norm separation problem. More precisely, we have

$$\epsilon = \sqrt{\frac{2(1 - \beta(t))}{\tilde{\gamma}(t)}}, \delta = \frac{K^2(K-1)n}{2e\tilde{\gamma}(t)q_{min}(t)},$$

where:

$$\beta = \langle \frac{\theta}{||\theta||_2}, \frac{d\theta}{dt}/||\frac{d\theta}{dt}||_2 \rangle$$

is the angle between $\theta$ and its corresponding gradient.

*Proof.* The training dynamics is given by gradient flow:

$$\frac{d\theta}{dt} = -\frac{\partial \mathcal{L}(\theta)}{\partial \theta}.$$

Then by the chain rule we have:

$$-\frac{d\mathcal{L}(\theta)}{dt} = -\frac{\partial \mathcal{L}}{\partial \theta}\frac{d\theta}{dt} = \left\|\frac{d\theta}{dt}\right\|^2. \tag{B.4}$$

It indicates that the loss function $\mathcal{L}$ is monotonically decreasing. If $\mathcal{L}(\theta(t_0)) < \log 2$, we have $\mathcal{L}(\theta(t)) < \log 2, \forall t > t_0$. On the other hand, note that

$$\mathcal{L}(\theta(t)) = \sum_{k=1}^{K}\sum_{i=1}^{n}\log(1 + \sum_{j \neq k}e^{-s_{k,i,j}(t)}) \geq \log(1 + \exp(-q_{min}(t))), \tag{B.5}$$

which gives us $q_{min}(t) > 0, \forall t > t_0$.

Let $g = \frac{d\theta}{dt}$, note that we can rewrite the ULPM objective function (B) as $\mathcal{L}(\theta) = \sum_{k=1}^{K}\sum_{i=1}^{n}\log(1 + \sum_{j \neq k}e^{-s_{k,i,j}})$. By the chain rule and the gradient flow equation we have

$$g = -\frac{\partial \mathcal{L}}{\partial \theta}\sum_{k=1}^{K}\sum_{i=1}^{n}\sum_{j \neq k}\frac{e^{-s_{k,i,j}}}{1 + \sum_{l \neq k}e^{-s_{k,i,l}}}g_{k,i,j},$$

where $g_{k,i,j}$ is the gradient of $s_{k,i,j}$, i.e. $g_{k,i,j} = \nabla_\theta s_{k,i,j}(\theta)$. Now let $\tilde{g}_{k,i,j} = g_{k,i,j}/q_{min}^{1/2} = \nabla_\theta s_{k,i,j}(\tilde{\theta})$ and construct $\lambda_{k,i,j} = \frac{\rho}{||g||_2}\frac{e^{-s_{k,i,j}}}{1 + \sum_{l \neq k}e^{-s_{k,i,l}}}$, we only need to show:

$$||\tilde{\theta} - \sum_{k=1}^{K}\sum_{i=1}^{n}\sum_{j \neq k}\lambda_{k,i,j}\tilde{g}_{k,i,j}||_2^2 \leq \frac{1 - \beta}{\tilde{\gamma}}, \tag{B.6}$$

$$\sum_{k=1}^{K}\sum_{i=1}^{n}\sum_{j \neq k}\lambda_{k,i,j}(s_{k,i,j}(\tilde{\theta}) - 1) \leq \frac{K^2(K-1)n}{2eq_{min}\tilde{\gamma}}. \tag{B.7}$$

To prove (B.6), we only need to compute (Recall that $\tilde{\theta} = \theta/q_{min}(\theta)^{1/2}$):

$$||\tilde{\theta} - \sum_{k=1}^{K}\sum_{i=1}^{n}\sum_{j \neq y_n}\lambda_{k,i,j}\tilde{g}_{k,i,j}||_2^2 = \frac{\rho^2}{q_{min}}||\frac{\theta}{||\theta||_2} - \frac{g}{||g||_2}||_2^2 = \frac{\rho^2}{q_{min}}(2 - 2\beta).$$

Note that:

$$\tilde{\gamma} = \frac{-\log(e^{\mathcal{L}(\theta)} - 1)}{\rho^2}, \quad \mathcal{L}(\theta) = \sum_{n=1}^{N}\log(1 + \sum_{j \neq y_n}e^{-s_{nj}}) \geq \log(1 + \exp(-q_{min})). \tag{B.8}$$

Then we have the following inequality:

$$\tilde{\gamma} \leq \frac{q_{min}}{\rho^2}. \tag{B.9}$$

Take this back into (B.8) we have (B.6) as desired.

To prove (B.7), first by our construction:

$$\sum_{k=1}^{K}\sum_{i=1}^{n}\sum_{j \neq k}\lambda_{k,i,j}(s_{k,i,j}(\tilde{\theta}) - 1) = \frac{\rho}{q_{min}||g||_2}\sum_{k=1}^{K}\sum_{i=1}^{n}\sum_{j \neq k}\frac{e^{-s_{k,i,j}}}{1 + \sum_{l \neq k}e^{-s_{k,i,l}}}(s_{k,i,j} - q_{min}).$$
$$\tag{B.10}$$

Note that $||g||_2 \geq \left\langle g, \frac{\theta}{||\theta||_2} \right\rangle = \frac{1}{\rho} \langle g, \theta \rangle$ and $\langle g_{k,i,j}, \theta \rangle = 2s_{k,i,j}$ since $s_{k,i,j} = \boldsymbol{w}_k^\top \boldsymbol{h}_{k,i} - \boldsymbol{w}_j^\top \boldsymbol{h}_{k,i}$, we have:

$$
\begin{aligned}
||g||_2 \geq \frac{1}{\rho} \langle g, \theta \rangle = & \frac{1}{\rho} \sum_{k=1}^{K} \sum_{i=1}^{n} \sum_{j \neq k} \frac{e^{-s_{k,i,j}}}{1 + \sum_{l \neq k} e^{-s_{k,i,l}}} \langle g_{k,i,j}, \theta \rangle \\
= & \frac{2}{\rho} \sum_{k=1}^{K} \sum_{i=1}^{n} \sum_{j \neq k} \frac{e^{-s_{k,i,j}}}{1 + \sum_{l \neq k} e^{-s_{k,i,l}}} s_{k,i,j} \\
\geq & \frac{2}{\rho} \frac{q_{min}}{K} \sum_{k=1}^{K} \sum_{i=1}^{n} \sum_{j \neq k} e^{-s_{k,i,j}} \quad \text{(since } s_{k,i,j} \geq q_{min} > 0 \text{ and } e^{-s_{k,i,l}} \leq 1) \\
\geq & \frac{2}{\rho} \frac{q_{min}}{K} e^{-q_{min}}.
\end{aligned}
$$

(B.11)

Take this inequality back into the (B.10) we have:

$$
\begin{aligned}
\sum_{k=1}^{K} \sum_{i=1}^{n} \sum_{j \neq k} \lambda_{k,i,j}(s_{k,i,j}(\tilde{\theta}) - 1) \leq & \frac{K\rho^2}{2q_{min}^2} \sum_{k=1}^{K} \sum_{i=1}^{n} \sum_{j \neq k} \frac{e^{q_{min} - s_{k,i,j}}}{1 + \sum_{l \neq k} e^{-s_{k,i,l}}} (s_{k,i,j} - q_{min}) \\
\leq & \frac{K\rho^2}{2q_{min}^2} \sum_{k=1}^{K} \sum_{i=1}^{n} \sum_{j \neq k} e^{q_{min} - s_{k,i,j}} (s_{k,i,j} - q_{min}) \\
\leq & \frac{K}{2q_{min}\tilde{\gamma}} \sum_{k=1}^{K} \sum_{i=1}^{n} \sum_{j \neq k} e^{q_{min} - s_{k,i,j}} (s_{k,i,j} - q_{min}) \\
\leq & \frac{K^2(K-1)n}{2eq_{min}\tilde{\gamma}}.
\end{aligned}
$$

Where the last inequality is obtained from the fact $xe^{-x} \leq \frac{1}{e}, \forall x > 0$, which can be proved by some elementary calculus. $\qquad \square$

Based on Lemma B.1, we have shown that the $(\boldsymbol{W}, \boldsymbol{H})$ will be a $(\epsilon, \delta)$-KKT point, if we can show $(\epsilon, \delta)$ converge to zero, then by Theorem A.1 we know the limit point will be along the direction of a KKT point. Ignoring the constant term, we only have to show how $\tilde{\gamma}(t), \beta(t)$ and $q_{min}(t)$ evolve along time. Now we provide the following lemmas to illustrate their dynamics. The first lemma aims at proving that the norms of parameter $\rho(t)$ and $\tilde{\gamma}(t)$ are monotonically increasing.

**Lemma B.2.** If there exist $t_0$ such that $\mathcal{L}(\theta(t_0)) < \log 2$, then for any $t > t_0$ we have:

$$
\frac{d\rho^2}{dt} > 0, \frac{d\tilde{\gamma}}{dt} \geq 0.
$$

(B.12)

*Proof.* We can disentangle the whole training dynamics into the following two parts:

- the radial part: $v := \hat{\theta}\hat{\theta}^\top \frac{d\theta}{dt}$,

- the tangent part: $u = (I - \hat{\theta}\hat{\theta}^\top)\frac{d\theta}{dt}$.

First analyze the radial part, by the chain rule: $||v||_2 = |\hat{\theta}^\top \frac{d\theta}{dt}| = |\frac{1}{\rho} \langle \theta, \frac{d\theta}{dt} \rangle| = |\frac{1}{\rho} \frac{1}{2} \frac{d\rho^2}{dt}|$. For $\frac{d\rho^2}{dt}$, we have the following equation:

$$
\frac{1}{2} \frac{d\rho^2}{dt} = \left\langle \theta, \frac{d\theta}{dt} \right\rangle = 2 \sum_{k=1}^{K} \sum_{i=1}^{n} \sum_{j \neq k} \frac{e^{-s_{k,i,j}}}{1 + \sum_{l \neq k} e^{-s_{k,i,l}}} s_{k,i,j},
$$

(B.13)

where the last equality holds by equation (B.11).Then when $t > t_0$, we have shown that $q_{min}(t) \geq 0$, combine this with the fact $s_{k,i,j} \geq q_{min}$ we obtain the first inequality in (B.12)

$$
\begin{aligned}
\frac{1}{2}\frac{d\rho^2}{dt} &= 2\sum_{k=1}^{K}\sum_{i=1}^{n}\sum_{j\neq k}\frac{e^{-s_{k,i,j}}}{1+\sum_{l\neq k}e^{-s_{k,i,l}}}s_{k,i,j} \\
&\geq 2\sum_{k=1}^{K}\sum_{i=1}^{n}\sum_{j\neq k}\frac{e^{-s_{k,i,j}}}{1+\sum_{l\neq k}e^{-s_{k,i,l}}}q_{min} \geq 0.
\end{aligned}
\tag{B.14}
$$

Next, we aim to prove the monotonicity of $\tilde{\gamma}(t)$, compute the derivative of $\tilde{\gamma}(t)$ we have:

$$
\tilde{\gamma} = \frac{-\log(e^{\mathcal{L}(\theta)}-1)}{\rho^2}, \quad \frac{d}{dt}\log\tilde{\gamma} = \frac{d}{dt}(\log(-\log(e^{\mathcal{L}(\theta)}-1))-2\log\rho),
\tag{B.15}
$$

$$
\frac{d}{dt}\log(-\log(e^{\mathcal{L}(\theta)}-1)) = \frac{1}{\log(e^{\mathcal{L}(\theta)}-1)}\frac{e^{\mathcal{L}(\theta)}}{e^{\mathcal{L}(\theta)}-1}\frac{d\mathcal{L}(\theta)}{dt} \geq -\frac{d\mathcal{L}(\theta)}{dt}\frac{1}{q_{min}}\frac{e^{\mathcal{L}(\theta)}}{e^{\mathcal{L}(\theta)}-1}.
\tag{B.16}
$$

Recall that we have:

$$
\begin{aligned}
\frac{1}{2}\frac{d\rho^2}{dt} &= 2\sum_{k=1}^{K}\sum_{i=1}^{n}\sum_{j\neq k}\frac{e^{-s_{k,i,j}}}{1+\sum_{l\neq k}e^{-s_{k,i,l}}}s_{k,i,j} \\
&\geq 2\sum_{k=1}^{K}\sum_{i=1}^{n}\frac{\sum_{j\neq k}e^{-s_{k,i,j}}}{1+\sum_{j\neq k}e^{-s_{k,i,j}}}q_{min} \\
&= \sum_{k=1}^{K}\sum_{i=1}^{n}\log(1+\sum_{j\neq k}e^{-s_{k,i,j}})\frac{1}{\log(1+\sum_{j\neq k}e^{-s_{k,i,j}})}\frac{\sum_{j\neq k}e^{-s_{k,i,j}}}{1+\sum_{j\neq k}e^{-s_{k,i,j}}}q_{min} \\
&\geq 2\sum_{k=1}^{K}\sum_{i=1}^{n}\log(1+\sum_{j\neq k}e^{-s_{k,i,j}})\frac{e^{\mathcal{L}(\theta)}-1}{\mathcal{L}(\theta)e^{\mathcal{L}(\theta)}}q_{min} \\
&= 2\frac{e^{\mathcal{L}(\theta)}-1}{e^{\mathcal{L}(\theta)}}q_{min}.
\end{aligned}
\tag{B.17}
$$

Then second last line is because the definition of the loss function $\mathcal{L}(\theta) = \sum_{k=1}^{K}\sum_{i=1}^{n}\log(1+\sum_{j\neq k}e^{-s_{k,i,j}}) \geq \log(1+\sum_{j\neq k}e^{-s_{k,i,j}})$ and the monotonicity of $\frac{e^x-1}{xe^x}$ (in fact, $\frac{d}{dx}\frac{e^x-1}{xe^x} = \frac{e^{-x}(x-e^x+1)}{x^2} \leq 0, \forall x > 0$).

As a result, we notice that

$$
\frac{1}{2}\frac{d}{dt}\log\tilde{\gamma}(t) \geq -\left(\frac{1}{2}\frac{d\rho^2}{dt}\right)^{-1}\frac{d\mathcal{L}}{dt} - \frac{d}{dt}\log\rho.
$$

At the same time, we notice that $\|v\|^2 = \frac{1}{\rho^2}\left(\frac{1}{2}\frac{d\rho^2}{dt}\right)^2 = \frac{1}{2}\frac{d\rho^2}{dt}\cdot\frac{d}{dt}\log\rho$ on the one hand, and by the chain rule:

$$
\frac{d}{dt}\hat{\theta} = \frac{1}{\rho^2}\left(\rho\frac{d\theta}{dt} - \frac{d\rho}{dt}\theta\right) = \frac{1}{\rho^2}\left(\rho\frac{d\theta}{dt} - \left(\hat{\theta}^\top\frac{d\theta}{dt}\right)\theta\right) = \frac{u}{\rho}.
$$

Combine this with the radial term:

$$
-\frac{d\mathcal{L}}{dt} = \left\|\frac{d\theta}{dt}\right\|^2 = \|v\|^2 + \|u\|^2 = \frac{1}{2}\frac{d\rho^2}{dt}\cdot\frac{d}{dt}\log\rho + \rho^2\left\|\frac{d\hat{\theta}}{dt}\right\|^2.
$$

Dividing $\frac{1}{2}\frac{d\rho^2}{dt}$ on both sides, we have

$$
-\frac{d\mathcal{L}}{dt}\cdot\left(\frac{1}{2}\frac{d\rho^2}{dt}\right)^{-1} = \frac{d}{dt}\log\rho + \left(\frac{d}{dt}\log\rho\right)^{-1}\left\|\frac{d\hat{\theta}}{dt}\right\|^2,
\tag{B.18}
$$

$$\frac{d}{dt} \log \rho + \left(\frac{d}{dt} \log \rho\right)^{-1} \left\|\frac{d\hat{\theta}}{dt}\right\|^2 \leq -\frac{d\mathcal{L}(\theta)}{dt} \frac{1}{q_{min}} \frac{e^{\mathcal{L}(\theta)}}{2(e^{\mathcal{L}(\theta)} - 1)}.$$

Now by equation (B.15) and (B.16) we obtain:

$$\frac{1}{2}\frac{d}{dt} \log \tilde{\gamma} \geq -\frac{d\mathcal{L}(\theta)}{dt} \frac{1}{q_{min}} \frac{e^{\mathcal{L}(\theta)}}{2\left(e^{\mathcal{L}(\theta)} - 1\right)} - \frac{d}{dt} \log \rho \geq \left(\frac{d}{dt} \log \rho\right)^{-1} \left\|\frac{d\hat{\theta}}{dt}\right\|^2. \tag{B.19}$$

By (B.14) we know the $\rho$ is monotonically increasing, we have $\frac{d}{dt} \log \rho > 0$ and then we get the second inequality in (B.12). $\qquad\square$

Lemma B.2 gives us the monotonicity of $\tilde{\gamma}$, note that since the loss function $\mathcal{L}(\theta(t_0)) < \log 2$, we have $\tilde{\gamma}(t) \geq \tilde{\gamma}(t_0) > 0$, then we can treat $\tilde{\gamma}(t)$ in Lemma B.1 as a positive constant. The remaining work is to show $q_{min}(t)$ grows to infinity and $\beta(t) \to 1$. To show $q_{min}(t) \to \infty$, it's equivalent to show $\mathcal{L}(t) \to 0$ and we have the following lemma:

**Lemma B.3.** If there exist $t_0$ such that $\mathcal{L}(\theta(t_0)) < \log 2$, then $\mathcal{L}(\theta(t)) \to 0$ and $q_{min}(\theta(t)) \to \infty$ as $t \to \infty$, moreover we have the convergence rate $\mathcal{L}(\theta(t)) = O(1/t)$

*Proof.* By (B.4) and (B.13), the evolution of loss function $\mathcal{L}(\theta)$ can be written as:

$$\frac{d\mathcal{L}(\theta)}{dt} = -\left\|\frac{d\theta}{dt}\right\|^2 \leq -\left\langle\frac{d\theta}{dt}, \frac{\theta}{\|\theta\|_2}\right\rangle^2 = -(2\sum_{k=1}^{K}\sum_{i=1}^{n}\sum_{j\neq k} \frac{e^{-s_{k,i,j}}}{1 + \sum_{l\neq k} e^{-s_{k,i,l}}} s_{k,i,j})^2.$$

Combine it with (B.5),(B.14) and (B.17) we have:

$$2\sum_{k=1}^{K}\sum_{i=1}^{n}\sum_{j\neq k} \frac{e^{-s_{k,i,j}}}{1 + \sum_{l\neq k} e^{-s_{k,i,l}}} s_{k,i,j} \geq 2\frac{e^{\mathcal{L}(\theta)} - 1}{e^{\mathcal{L}(\theta)}} q_{min} \geq -2\frac{e^{\mathcal{L}(\theta)} - 1}{e^{\mathcal{L}(\theta)}} \log(e^{\mathcal{L}(\theta)} - 1),$$

which indicates:

$$\frac{d\mathcal{L}(\theta(t))}{dt} \leq -4(\frac{e^{\mathcal{L}(\theta(t))} - 1}{e^{\mathcal{L}(\theta(t))}} \log(e^{\mathcal{L}(\theta(t))} - 1))^2. \tag{B.20}$$

Since $0 < \mathcal{L}(\theta(t)) < \mathcal{L}(\theta(t_0)) < \log 2$, note that:

$$\frac{d}{dx}(\frac{e^x - 1}{x}) > 0, \lim_{x\to 0} \frac{e^x - 1}{x} = 1,$$

which implies:

$$1 < \frac{e^{\mathcal{L}(\theta(t))} - 1}{\mathcal{L}(\theta(t))} < \frac{1}{\log(2)}, \tag{B.21}$$

On the other hand, we can find that:

$$\log(e^{\mathcal{L}(\theta(t))} - 1) < \log(e^{\mathcal{L}(\theta(t_0))} - 1) < 0, \quad 1 < e^{\mathcal{L}(\theta(t))} < 2 \tag{B.22}$$

Combine equation (B.21) and (B.22) together, one can conclude that there exist a constant $C > 0$ such that for $t > t_0$:

$$\frac{d\mathcal{L}(\theta(t))}{dt} \leq -C(\mathcal{L}(\theta(t)))^2$$

It further implies that:

$$C < \frac{d}{dt}(\frac{1}{\mathcal{L}(\theta(t))}),$$

then integral on both side we have:

$$C(t - t_0) < \frac{1}{\mathcal{L}(\theta(t))} - \frac{1}{\mathcal{L}(\theta(t_0))},$$

and

$$\mathcal{L}(\theta(t)) = O(1/t).$$

Thus we must have $\mathcal{L}(\theta(t)) \to 0$ and combine this with $q_{min} \geq -\log(e^{\mathcal{L}(\theta(t))} - 1)$ we know $q_{min}(\theta(t)) \to \infty$ as desired. $\qquad\square$

To bound $\beta(t)$, we first need a useful lemma to bound the changes of the direction of $\theta$.

**Lemma B.4.** If there exist $t_0$ such that $\mathcal{L}(\theta(t_0)) < \log 2$, then for any $t > t_0$

$$\left\|\frac{d\hat{\theta}}{dt}\right\| \leq \frac{1}{\tilde{\gamma}(t_0)}\frac{d}{dt}\log\rho.$$

*Proof.* First we know that:

$$\left\|\frac{d\hat{\theta}}{dt}\right\| = \frac{1}{\rho}\left\|(I - \hat{\theta}\hat{\theta}^\top)\frac{d\theta}{dt}\right\| \leq \frac{1}{\rho}\left\|\frac{d\theta}{dt}\right\|,$$

$$\left\|\frac{d\theta}{dt}\right\| = \left\|\sum_{k=1}^K\sum_{i=1}^n\sum_{j\neq k}\frac{e^{-s_{k,i,j}}}{1+\sum_{l\neq k}e^{-s_{k,i,l}}}g_{k,i,j}\right\| \leq \sum_{k=1}^K\sum_{i=1}^n\sum_{j\neq k}\frac{e^{-s_{k,i,j}}}{1+\sum_{l\neq k}e^{-s_{k,i,l}}}\|g_{k,i,j}\|.$$

(B.23)

Recall our $g_{k,i,j} = \frac{\partial s_{k,i,j}}{\partial\theta}$ and $s_{k,i,j} = \boldsymbol{w}_k^\top\boldsymbol{h}_{k,i} - \boldsymbol{w}_j^\top\boldsymbol{h}_{k,i}$, we can find that $\|g_{k,i,j}\| \leq 2\rho$. On the other hand, Combine it with (B.13) and (B.9) we have:

$$\left\|\frac{d\hat{\theta}}{dt}\right\| \leq \frac{1}{\rho}\left\|\frac{d\theta}{dt}\right\| \leq \frac{1}{2q_{min}}\frac{d\rho^2}{dt} = \frac{\rho^2}{q_{min}}\frac{d}{dt}\log\rho \leq \frac{1}{\tilde{\gamma}}\frac{d}{dt}\log\rho \leq \frac{1}{\tilde{\gamma}(t_0)}\frac{d}{dt}\log\rho$$

as desired. Where the second inequality holds by multiple $\frac{\rho}{q_{min}}$ on the right hand of (B.23) and the formula of $\frac{d\rho^2}{dt}$ in equation (B.13), and the last inequality holds since we have shown that $\tilde{\gamma}(t)$ is monotonically increasing in (B.2) and $\tilde{\gamma}(t_0) > 0$ since $\mathcal{L}(t_0) < \log 2$. $\square$

Let's turn back to $\beta$, though we can not show directly that it increase to one, we can find a sequence of time $\{t_m\}$ for each limit point such that $\beta(t_m) \to 1$

**Lemma B.5.** If there exist $t_0$ such that $\mathcal{L}(\theta(t_0)) < \log 2$, then for every limit point $\bar{\theta}$ of $\{\hat{\theta}(t) : t \geq 0\}$, there exists a sequence of time $\{t_m > 0 : m \in \mathbb{N}\}$ such that $t_m \to \infty, \hat{\theta}(t_m) \to \bar{\theta}$, and $\beta(t_m) \to 1$.

*Proof.* Recall that in equation (B.19) we have shown that:

$$\frac{d}{dt}\log\tilde{\gamma} \geq 2\left(\frac{d}{dt}\log\rho\right)^{-1}\left\|\frac{d\hat{\theta}}{dt}\right\|^2.$$

(B.24)

Since $\frac{d}{dt}\log\rho = \frac{1}{\rho}\frac{d\rho}{dt} = \frac{1}{2\rho^2}\frac{d\rho^2}{dt} = \frac{1}{\rho^2}\langle\theta,\frac{d\theta}{dt}\rangle$ and:

$$\frac{d\hat{\theta}}{dt} = \frac{d}{dt}\frac{\theta}{\|\theta\|} = \frac{1}{\rho^2}(\rho\frac{d\theta}{dt} - \frac{1}{\rho}\theta\theta^\top\frac{d\theta}{dt}) = \frac{1}{\rho}(I - \hat{\theta}\hat{\theta}^\top)\frac{d\theta}{dt}.$$

Plug them into (B.24) we have:

$$\frac{d}{dt}\log\tilde{\gamma} \geq 2\frac{\left\|\frac{d\theta}{dt}\right\|^2 - \left\langle\hat{\theta},\frac{d\theta}{dt}\right\rangle^2}{\left\langle\hat{\theta},\frac{d\theta}{dt}\right\rangle^2}\frac{d}{dt}\log\rho = 2(\beta^{-2} - 1)\frac{d}{dt}\log\rho.$$

For any $t_2 > t_1 > t_0$, integrate both sides from time $t_2$ to $t_1$ we have: $\log\tilde{\gamma}(t_2) - \log\tilde{\gamma}(t_1) \geq 2\int_{t_1}^{t_2}(\beta(t)^{-2} - 1)\frac{d}{dt}\log\rho dt$. By the continuity of $\beta$ we know there exist a time $t^*$ such that:

$$\begin{aligned}\log\tilde{\gamma}(t_2) - \log\tilde{\gamma}(t_1) &\geq 2\int_{t_1}^{t_2}(\beta(t)^{-2} - 1)\frac{d}{dt}\log\rho dt \\ &= 2(\beta(t^*)^{-2} - 1)\int_{t_1}^{t_2}\frac{d}{dt}\log\rho dt \\ &= 2(\beta(t^*)^{-2} - 1)(\log\rho(t_2) - \log\rho(t_1)).\end{aligned}$$

(B.25)

By (B.9) we know that $\tilde{\gamma} \leq \frac{q_{min}}{\rho^2}$ and the right hand is bounded, and the $\tilde{\gamma}$ is monotonically increasing, then there exist $\tilde{\gamma}_\infty$ such that $\tilde{\gamma}(t) \uparrow \tilde{\gamma}_\infty$.

Now we are ready to construct the sequence of $t_m$, first take a sequence of $\{\epsilon_m > 0, m \in \mathbb{N}\}$ such that $\epsilon_m \to 0$. We construct $t_m$ by induction, suppose we have already find $t_1 < t_2 < \cdots < t_{m-1}$ satisfy our requirement, since $\bar{\theta}$ is a limit point of $\{\hat{\theta}(t) : t > 0\}$, then we can find a time $s_m$ such that:

$$\|\hat{\theta}(s_m) - \bar{\theta}\| \leq \epsilon_m, \quad \log \frac{\tilde{\gamma}_\infty}{\tilde{\gamma}(s_m)} \leq \epsilon_m^3.$$

By the monotonicity and continuity of $\rho$ we can find a time $s'_m$ such that $\log \rho(s'_m) - \log \rho(s_m) \leq \epsilon_m$. Take $t_2 = s'_m, t_1 = s_m$ in (B.25), there exist a time $t_m$ such that:

$$2(\beta(t_m)^{-2} - 1) \leq \frac{\log \tilde{\gamma}(t_2) - \log \tilde{\gamma}(t_1)}{\log \rho(t_2) - \log \rho(t_1)} \leq \epsilon_m^2 \tag{B.26}$$

on the other hand, by Lemma B.4 we have:

$$\begin{aligned}
\|\hat{\theta}(t_m) - \bar{\theta}\| &\leq \|\hat{\theta}(s_m) - \bar{\theta}\| + \|\hat{\theta}(s_m) - \hat{\theta}(t_m)\| \\
&\leq \epsilon_m + \frac{1}{\tilde{\gamma}(t_0)}(\log \rho(t_m) - \log \rho(s_m)) \leq (1 + \frac{1}{\tilde{\gamma}(t_0)})\epsilon_m.
\end{aligned} \tag{B.27}$$

Note that $\langle \theta, \frac{d\theta}{dt} \rangle > 0$, then by definition we know $\beta > 0$. Combine equations (B.26) and (B.27) we have $\beta(t_m) \to 1$ and $\hat{\theta}(t_m) \to \bar{\theta}$ as desired. $\qquad \square$

Now we are ready to prove Theorem 3.2:

*Proof of Theorem 3.2.* By Lemma B.1, we know that once $t > t_0$, $(\boldsymbol{W}(t), \boldsymbol{H}(t))/q_{min}(\boldsymbol{W}(t), \boldsymbol{H}(t))$ is an $(\sqrt{\frac{2(1-\beta(t))}{\tilde{\gamma}(t)}}, \frac{K^2(K-1)n}{2\tilde{\gamma}(t)q_{min}(t)})$ -approximate KKT point. We have shown that $\tilde{\gamma}(t) > \tilde{\gamma}(t_0) > 0$ in Lemma B.2, $q_{min} \to \infty$ in Lemma B.3 and from Lemma B.5 we know for any limit point $(\bar{\boldsymbol{W}}, \bar{\boldsymbol{H}})$ of $\{(\hat{\boldsymbol{H}}(t), \hat{\boldsymbol{W}}(t)) := (\frac{\boldsymbol{H}(t)}{\sqrt{\|\boldsymbol{W}(t)\|_F^2 + \|\boldsymbol{H}(t)\|_F^2}}, \frac{\boldsymbol{W}(t)}{\sqrt{\|\boldsymbol{W}(t)\|_F^2 + \|\boldsymbol{H}(t)\|_F^2}})\}$, there exists a sequence of time $\{t_m > 0 : m \in \mathbb{N}\}$ such that $t_m \to \infty, \beta(t_m) \to 1$ and $(\hat{\boldsymbol{H}}(t_m), \hat{\boldsymbol{W}}(t_m)) \to (\bar{\boldsymbol{W}}, \bar{\boldsymbol{H}})$. Then $(\bar{\boldsymbol{W}}, \bar{\boldsymbol{H}})$ is along the direction of a limit point of a sequence of $(\epsilon, \delta)$-approximate KKT point with $\epsilon, \delta \to 0$. On the other hand, we can verify that the problem (3.1) satisfies MCFQ (A.3) by simply setting $v = \theta$, then:

$$\langle \nabla s_{k,i,j}, \theta \rangle = 2s_{k,i,j} \geq 0.$$

Now by Theorem A.1 we know $(\bar{\boldsymbol{W}}, \bar{\boldsymbol{H}})$ is along the direction of a KKT point of problem (3.1) $\quad \square$

Theorem 3.2 characterize the convergent behaviour of gradient flow, under separable conditions the limit point is along the direction of a KKT point of (3.1), next we show that the global minimum of (3.1) must satisfy neural collapse conditions by proving Corollary 3.1:

*Proof of Corollary 3.1.* Since we have shown that the problem (3.1) satisfy MCFQ, then the KKT conditions are necessary for global optimality, we only need to show the global optimum satisfies neural collapse conditions. First the constraints in (3.1) can be transformed to be a single constraint by the definition of margin:

$$\forall k \neq j \in [K], i \in [n], \quad \boldsymbol{w}_k^\top \boldsymbol{h}_{k,i} - \boldsymbol{w}_j^\top \boldsymbol{h}_{k,i} \geq 1. \Leftrightarrow q_{\min}(\boldsymbol{W}, \boldsymbol{H}) \geq 1. \tag{B.28}$$

Note that the margin is homogeneous:

$$q_{\min}(\alpha\boldsymbol{W}, \alpha\boldsymbol{H}) = \alpha^2 q_{\min}(\boldsymbol{W}, \boldsymbol{H}), \forall \alpha \in \mathbb{R}.$$

Then for any point $(\boldsymbol{W}, \boldsymbol{H})$ satisfies $q_{\min}(\boldsymbol{W}, \boldsymbol{H}) > 0$, after an appropriate scaling $\alpha$, $(\alpha\boldsymbol{W}, \alpha\boldsymbol{H}), \forall \alpha^2 \geq 1/q_{\min}(\boldsymbol{W}, \boldsymbol{H})$ is feasible for (3.1). Take optimum among all scaling factor $\alpha$ we know the minimum norm is attained if and only if $\alpha^2 = 1/q_{\min}(\boldsymbol{W}, \boldsymbol{H})$. And the optimum norm is:

$$\frac{1}{2}\|\alpha\boldsymbol{W}\|_F^2 + \frac{1}{2}\|\alpha\boldsymbol{H}\|_F^2 = \frac{1}{2q_{\min}(\boldsymbol{W}, \boldsymbol{H})}(\|\boldsymbol{W}\|_F^2 + \|\boldsymbol{H}\|_F^2).$$

Then by Theorem 3.1 we have:

$$\frac{1}{2q_{\min}(\boldsymbol{W},\boldsymbol{H})}(\|\boldsymbol{W}\|_F^2 + \|\boldsymbol{H}\|_F^2) \geq 2(K-1)\sqrt{n}.$$

And the global optimum is attained only when $(\boldsymbol{W},\boldsymbol{H})$ satisfies neural collapse conditions □

## C  OMITTED PROOFS FROM SECTION 3.2

To begin with, let's finish the computation in the motivating example (Example 3.1)

*Proof in Example 3.1.* Consider the case where $K = 4, n = 1$, let $(\boldsymbol{W},\boldsymbol{H})$ be the following point:

$$\boldsymbol{W} = \boldsymbol{H} = \begin{bmatrix} 1 & -1 & 0 & 0 \\ -1 & 1 & 0 & 0 \\ 0 & 0 & 1 & -1 \\ 0 & 0 & -1 & 1 \end{bmatrix}.$$

One can easily verify that this $(\boldsymbol{W},\boldsymbol{H})$ enables our model to classify all of the features perfectly. Further more, we can show it is along the direction of a KKT point of the minimum-norm separation problem (3.1) by construct the Lagrangian multiplier $\Lambda = (\lambda_{ij})_{i,j=1}^K$ as following:

$$\Lambda = \begin{bmatrix} 0 & 0 & \frac{1}{2} & \frac{1}{2} \\ 0 & 0 & \frac{1}{2} & \frac{1}{2} \\ \frac{1}{2} & \frac{1}{2} & 0 & 0 \\ \frac{1}{2} & \frac{1}{2} & 0 & 0 \end{bmatrix}.$$

To see this, just write down the corresponding Lagrangian (note that to make it to be a true KKT point of (3.1), one needs to multiple $1/\sqrt{2}$ on $\boldsymbol{W}, \boldsymbol{H}$):

$$\mathcal{L}(\boldsymbol{W},\boldsymbol{H},\Lambda) = \frac{1}{4}\|\boldsymbol{W}\|_F^2 + \frac{1}{4}\|\boldsymbol{H}\|_F^2 - \sum_{i=1}^4\sum_{j\neq i}\lambda_{i,j}(\frac{1}{2}\boldsymbol{w}_i\boldsymbol{h}_i - \frac{1}{2}\boldsymbol{w}_j\boldsymbol{h}_i - 1).$$

Simply take derivatives for $\boldsymbol{W}, \boldsymbol{H}$ and $\Lambda$ we can find it satisfies KKT conditions. On the other hand, the gradient of $(\boldsymbol{W},\boldsymbol{H})$ is

$$\nabla_{\boldsymbol{W}}\mathcal{L}(\boldsymbol{W},\boldsymbol{H}) = \nabla_{\boldsymbol{H}}\mathcal{L}(\boldsymbol{W},\boldsymbol{H}) = -\frac{2+2e^{-2}}{2+2e^{-2}+2e^2}\begin{bmatrix} 1 & -1 & 0 & 0 \\ -1 & 1 & 0 & 0 \\ 0 & 0 & 1 & -1 \\ 0 & 0 & -1 & 1 \end{bmatrix}.$$

We can find that the directions of gradient and the parameter align with each other (i.e., $\boldsymbol{W}$ is parallel to $\nabla_{\boldsymbol{W}}\mathcal{L}(\boldsymbol{W},\boldsymbol{H})$, and $\boldsymbol{H}$ is parallel to $\nabla_{\boldsymbol{H}}\mathcal{L}(\boldsymbol{W},\boldsymbol{H})$), which implies that simple gradient descent may gets stuck in this direction and only grows the parameters' norm. However, if we construct:

$$\boldsymbol{W}' = \sqrt{\frac{1}{1+2\alpha^2}}\begin{bmatrix} 1+\alpha & -1+\alpha & \alpha & \alpha \\ -1+\alpha & 1+\alpha & \alpha & \alpha \\ -\alpha & -\alpha & 1-\alpha & -1-\alpha \\ -\alpha & -\alpha & -1-\alpha & 1-\alpha \end{bmatrix},$$

$$\boldsymbol{H}' = \sqrt{\frac{1}{1+2\alpha^2}}\begin{bmatrix} 1+\alpha & -1+\alpha & -\alpha & -\alpha \\ -1+\alpha & 1+\alpha & -\alpha & -\alpha \\ \alpha & \alpha & 1-\alpha & -1-\alpha \\ \alpha & \alpha & -1-\alpha & 1-\alpha \end{bmatrix}.$$

Note that $\|\boldsymbol{W}'\|_F = \|\boldsymbol{W}\|_F, \|\boldsymbol{H}'\|_F = \|\boldsymbol{H}\|_F$ and $\|\boldsymbol{W}' - \boldsymbol{W}\|_F^2 + \|\boldsymbol{H}' - \boldsymbol{H}\|_F^2 \to 0$ as $\alpha \to 0$. First we can compute:

$$\boldsymbol{W}'\boldsymbol{H}' = \frac{1}{1+2\alpha^2}\begin{bmatrix} 2+4\alpha^2 & 4\alpha^2-2 & -4\alpha^2 & -4\alpha^2 \\ 4\alpha^2-2 & 2+4\alpha^2 & -4\alpha^2 & -4\alpha^2 \\ -4\alpha^2 & -4\alpha^2 & 2+4\alpha^2 & 4\alpha^2-2 \\ -4\alpha^2 & -4\alpha^2 & 4\alpha^2-2 & 2+4\alpha^2 \end{bmatrix},$$

and:

$$\mathcal{L}(\boldsymbol{W}', \boldsymbol{H}') = -4 \log \frac{e^2}{e^2 + e^{2\frac{2\alpha^2-1}{1+2\alpha^2}} + 2e^{-2\frac{2\alpha^2}{1+2\alpha^2}}}.$$

Our aim is to show that for any $\epsilon > 0$, there exist $\alpha$ such that $|\alpha| < \epsilon$ and $\mathcal{L}(\boldsymbol{W}', \boldsymbol{H}') < \mathcal{L}(\boldsymbol{W}, \boldsymbol{H})$. By the formulation of $\mathcal{L}(\boldsymbol{W}', \boldsymbol{H}')$, it's sufficient to show that $f(\alpha) \triangleq e^{2\frac{2\alpha^2-1}{1+2\alpha^2}} + 2e^{-2\frac{2\alpha^2}{1+2\alpha^2}} < f(0)$. Then take the derivative of $f(\alpha)$ we have:

$$f'(\alpha) = e^{\frac{4\alpha^2-2}{1+2\alpha^2}} \left( \frac{8\alpha}{1+2\alpha^2} - \frac{8\alpha\left(2\alpha^2-1\right)}{\left(1+2\alpha^2\right)^2} \right) + 2e^{-\frac{4\alpha^2}{1+2\alpha^2}} \left( \frac{16\alpha^3}{\left(1+2\alpha^2\right)^2} - \frac{8\alpha}{1+2\alpha^2} \right),$$

$$\begin{aligned}
f''(\alpha) = {} & e^{\frac{4\alpha^2-2}{1+2\alpha^2}} \left( \frac{8\alpha}{1+2\alpha^2} - \frac{8\alpha\left(2\alpha^2-1\right)}{\left(1+2\alpha^2\right)^2} \right)^2 + 2e^{-\frac{4\alpha^2}{1+2\alpha^2}} \left( \frac{16\alpha^3}{\left(1+2\alpha^2\right)^2} - \frac{8\alpha}{1+2\alpha^2} \right)^2 \\
& + e^{\frac{4\alpha^2-2}{1+2\alpha^2}} \left( -\frac{64\alpha^2}{\left(1+2\alpha^2\right)^2} + \frac{64\left(2\alpha^2-1\right)\alpha^2}{\left(1+2\alpha^2\right)^3} + \frac{8}{1+2\alpha^2} - \frac{8\left(2\alpha^2-1\right)}{\left(1+2\alpha^2\right)^2} \right) \\
& + 2e^{-\frac{4\alpha^2}{1+2\alpha^2}} \left( \frac{80\alpha^2}{\left(1+2\alpha^2\right)^2} - \frac{8}{1+2\alpha^2} - \frac{128\alpha^4}{\left(1+2\alpha^2\right)^3} \right).
\end{aligned}$$

Now we can find that $f'(0) = 0$ and $f''(0) = 16(\frac{1}{e^2} - 1) < 0$. Since the function $f(\alpha)$ is continuously twice differentiable, we can conclude that for any $\epsilon > 0$, we can choose appropriate $\alpha$ such that:

$$||\boldsymbol{W}'||_F^2 = ||\boldsymbol{W}||_F^2, ||\boldsymbol{H}'||_F^2 = ||\boldsymbol{H}||_F^2,$$
$$||\boldsymbol{W}' - \boldsymbol{W}||_F^2 + ||\boldsymbol{H}' - \boldsymbol{H}||_F^2 < \epsilon, \mathcal{L}(\boldsymbol{W}', \boldsymbol{H}') < \mathcal{L}(\boldsymbol{W}, \boldsymbol{H}).$$

$\square$

Now we prove Theorem 3.3 by some similar strategies as used in proving Theorem 3.1.

*Proof of Theorem 3.3.* Again we rewrite the ULPM objective by introducing $s_{k,i,j} = \boldsymbol{w}_k^\top \boldsymbol{h}_{k,i} - \boldsymbol{w}_j^\top \boldsymbol{h}_{k,i}$:

$$\mathcal{L}(\boldsymbol{W}, \boldsymbol{H}) = \sum_{k=1}^{K} \sum_{i=1}^{n} \log(1 + \sum_{j \neq k} \exp(-s_{k,i,j})).$$

In addition, we can find that centralizing $\boldsymbol{w}_i$ does not change the value of $s_{k,i,j}$. Let $\tilde{\boldsymbol{w}}_i = \boldsymbol{w}_i - \frac{1}{K} \sum_{k=1}^{K} \boldsymbol{w}_k, \forall i \in [K]$, then $s_{k,i,j} = \boldsymbol{w}_k^\top \boldsymbol{h}_{k,i} - \boldsymbol{w}_j^\top \boldsymbol{h}_{k,i} = \tilde{\boldsymbol{w}}_k^\top \boldsymbol{h}_{k,i} - \tilde{\boldsymbol{w}}_j^\top \boldsymbol{h}_{k,i}$ and $\sum_{i=1}^{K} \tilde{\boldsymbol{w}}_i = 0$. First by the strict convexity of $e^x$ and Jensen Inequality:

$$\begin{aligned}
\mathcal{L}(\boldsymbol{W}, \boldsymbol{H}) & \geq \sum_{k=1}^{K} \sum_{i=1}^{n} \log(1 + (K-1) \exp(\frac{1}{K-1} \sum_{j \neq k} -s_{k,i,j})) \\
& = \sum_{k=1}^{K} \sum_{i=1}^{n} \log(1 + (K-1) \exp(-\frac{K\tilde{\boldsymbol{w}}_k^\top \boldsymbol{h}_{k,i}}{K-1})).
\end{aligned} \tag{C.1}$$

Where the last equality is obtained from:

$$\sum_{j \neq k} s_{k,i,j} = \sum_{j \neq k} \tilde{\boldsymbol{w}}_k^\top \boldsymbol{h}_{k,i} - \tilde{\boldsymbol{w}}_j^\top \boldsymbol{h}_{k,i} = (K-1)\tilde{\boldsymbol{w}}_k^\top \boldsymbol{h}_{k,i} - \sum_{j \neq k} \tilde{\boldsymbol{w}}_j^\top \boldsymbol{h}_{k,i} = K\tilde{\boldsymbol{w}}_k^\top \boldsymbol{h}_{k,i}.$$

Now again by the strict convexity of $\log(1 + (K-1)\exp(-x))$ and Jensen inequality, we have:

$$
\begin{aligned}
\mathcal{L}(\boldsymbol{W}, \boldsymbol{H}) &\geq \sum_{k=1}^{K} \sum_{i=1}^{n} \log(1 + (K-1)\exp(-\frac{K\tilde{\boldsymbol{w}}_k^\top \boldsymbol{h}_{k,i}}{K-1})) \\
&\geq nK \log(1 + (K-1)\exp(-\frac{1}{n(K-1)} \sum_{k=1}^{K} \sum_{i=1}^{n} \tilde{\boldsymbol{w}}_k^\top \boldsymbol{h}_{k,i})) \\
&\geq nK \log(1 + (K-1)\exp(-\frac{1}{2n(K-1)} \sum_{k=1}^{K} \sum_{i=1}^{n} \frac{1}{\sqrt{n}} \|\tilde{\boldsymbol{w}}_k\|^2 + \sqrt{n}\|\boldsymbol{h}_{k,i}\|^2)) \\
&\geq nK \log(1 + (K-1)\exp(-\frac{1}{2\sqrt{n}(K-1)} (\|\boldsymbol{W}\|_F^2 + \|\boldsymbol{H}\|_F^2))).
\end{aligned}
\tag{C.2}
$$

Where the last inequality holds since $\|\boldsymbol{W}\|_F^2 = \sum_{k=1}^{K} \|\boldsymbol{w}_k\|^2 \geq \sum_{k=1}^{K} \|\boldsymbol{w}_k\|^2 - \frac{1}{K}\|\sum_{k=1}^{K} \boldsymbol{w}_k\|^2 = \sum_{k=1}^{K} \|\tilde{\boldsymbol{w}}_k\|^2$.

When all of the above inequality reduce to equality, we must have:

1. $\sum_{i=1}^{K} \boldsymbol{w}_i = 0, \tilde{\boldsymbol{w}}_i = \boldsymbol{w}_i$ (the last inequality in (C.2))

2. $\boldsymbol{w}_k = \sqrt{n}\boldsymbol{h}_{k,i}, \forall i \in [n]$ (the third inequality in (C.2))

3. $\|\boldsymbol{w}_k\| = \|\boldsymbol{w}_{k'}\|, \|\boldsymbol{h}_{k,i}\| = \|\boldsymbol{h}_{k',j}\|, \forall k, k' \in [K], i, j \in [n]$ (the second inequality in (C.2))

4. $s_{k,i,j} = \boldsymbol{w}_k^\top \boldsymbol{h}_{k,i} - \boldsymbol{w}_j^\top \boldsymbol{h}_{k,i} = \frac{K}{K-1}\boldsymbol{w}_k^\top \boldsymbol{h}_{k,i}, \forall k, j \in [K], i \in [n]$ (the first inequality in (C.1))

These four conditions are exactly equivalent to neural collapse conditions and $\|\boldsymbol{W}\|_F = \|\boldsymbol{H}\|_F$ □

The global optimality is not enough to illustrate how does gradient flow converges to neural collapse since there may exist some bad local minimum. We will provide the following second-order analysis to eliminate spurious local minimum. First, define the cross-entropy loss on a matrix $\boldsymbol{Z} \in \mathbb{R}^{K \times nK}$:

$$
L(\boldsymbol{Z}) = \sum_{k=1}^{K} \sum_{i=1}^{n} -\log \frac{e^{z_{k,i,j}}}{\sum_{l=1}^{K} e^{z_{k,i,l}}},
$$

where $z_{k,i,j}$ denote the $j$-th row and $[(k-1)K + i]$-th column elements of $\boldsymbol{Z}$. Then we have $\mathcal{L}(\boldsymbol{W}, \boldsymbol{H}) = L(\boldsymbol{W}\boldsymbol{H})$. Now compute the gradient of $L(Z)$ to each element:

$$
\begin{aligned}
\frac{\partial L(\boldsymbol{Z})}{\partial z_{k,i,k}} &= -1 + \frac{z_{k,i,k}}{\sum_{l=1}^{K} e^{z_{k,i,l}}} \\
\frac{\partial L(\boldsymbol{Z})}{\partial z_{k,i,j}} &= \frac{z_{k,i,j}}{\sum_{l=1}^{K} e^{z_{k,i,l}}}, \forall k \neq j.
\end{aligned}
$$

If $u \in \mathbb{R}^K$ satisfies $u^\top \nabla L(Z) = 0$, denote $u_p$ as the maximum element of $u$, then we have:

$$
\begin{aligned}
0 = u_p \frac{\partial L(\boldsymbol{Z})}{\partial z_{p,i,p}} + \sum_{q \neq p} u_q \frac{\partial L(\boldsymbol{Z})}{\partial z_{p,i,q}} &= u_p(-1 + \frac{z_{p,i,p}}{\sum_{l=1}^{K} e^{z_{p,i,l}}}) + \sum_{q \neq p} u_q \frac{z_{p,i,q}}{\sum_{l=1}^{K} e^{z_{p,i,l}}} \\
&= -\sum_{q \neq p} (u_p - u_q) \frac{z_{p,i,q}}{\sum_{l=1}^{K} e^{z_{p,i,l}}} \leq 0.
\end{aligned}
\tag{C.3}
$$

Where the last inequality holds if and only if $u_q = u_p, \forall q \in [K]$. Which indicates that the rank of $\nabla L(Z)$ is $K-1$ and $u^\top \nabla L(Z) = 0 \Leftrightarrow u = \mathbf{1}$. Now we are ready to prove Theorem 3.4 in the main body.

*Proof of Theorem 3.4.* First compute the gradient of ULPM objective (B), by the chain rule we have:

$$
\nabla_{\boldsymbol{W}} \mathcal{L}(\boldsymbol{W}, \boldsymbol{H}) = \nabla L(\boldsymbol{W}\boldsymbol{H})\boldsymbol{H}^\top, \nabla_{\boldsymbol{H}} \mathcal{L}(\boldsymbol{W}, \boldsymbol{H}) = \boldsymbol{W}^\top \nabla L(\boldsymbol{W}\boldsymbol{H}).
$$

If there exist a vector $(\Delta \boldsymbol{W}, \Delta \boldsymbol{H}) \in \mathcal{T}(\boldsymbol{W}, \boldsymbol{H})$ such that $\langle \nabla_{\boldsymbol{W}} \mathcal{L}(\boldsymbol{W}, \boldsymbol{H}), \Delta \boldsymbol{W} \rangle + \langle \nabla_{\boldsymbol{H}} \mathcal{L}(\boldsymbol{W}, \boldsymbol{H}), \Delta \boldsymbol{H} \rangle \neq 0$, moreover we can assume $\langle \nabla_{\boldsymbol{W}} \mathcal{L}(\boldsymbol{W}, \boldsymbol{H}), \Delta \boldsymbol{W} \rangle + \langle \nabla_{\boldsymbol{H}} \mathcal{L}(\boldsymbol{W}, \boldsymbol{H}), \Delta \boldsymbol{H} \rangle < 0$ since we can take the negative direction if the formula is greater than zero, then by Taylor expansion:

$$\mathcal{L}(\boldsymbol{W} + \delta \Delta \boldsymbol{W}, \boldsymbol{H} + \delta \Delta \boldsymbol{H}) = \mathcal{L}(\boldsymbol{W}, \boldsymbol{H}) + \delta \langle \nabla_{\boldsymbol{W}} \mathcal{L}(\boldsymbol{W}, \boldsymbol{H}), \Delta \boldsymbol{W} \rangle + \delta \langle \nabla_{\boldsymbol{H}} \mathcal{L}(\boldsymbol{W}, \boldsymbol{H}), \Delta \boldsymbol{H} \rangle + O(\delta^2),$$

we know that $(\Delta \boldsymbol{W}, \Delta \boldsymbol{H})$ satisfies our requirement.

Now let's discuss the case when:

$$\langle \nabla_{\boldsymbol{W}} \mathcal{L}(\boldsymbol{W}, \boldsymbol{H}), \Delta \boldsymbol{W} \rangle + \langle \nabla_{\boldsymbol{H}} \mathcal{L}(\boldsymbol{W}, \boldsymbol{H}), \Delta \boldsymbol{H} \rangle = 0, \forall (\Delta \boldsymbol{W}, \Delta \boldsymbol{H}) \in \mathcal{T}(\boldsymbol{W}, \boldsymbol{H}),$$

by definition of $\mathcal{T}(\boldsymbol{W}, \boldsymbol{H})$, it contains all vectors that are orthogonal to $(\boldsymbol{W}, \boldsymbol{H})$, so $(\nabla_{\boldsymbol{W}} \mathcal{L}(\boldsymbol{W}, \boldsymbol{H}), \nabla_{\boldsymbol{H}} \mathcal{L}(\boldsymbol{W}, \boldsymbol{H}))$ is parallel to $(\boldsymbol{W}, \boldsymbol{H})$, that is, there exist $\lambda$ such that:

$$\nabla L(\boldsymbol{W} \boldsymbol{H}) \boldsymbol{H}^\top = \lambda \boldsymbol{W}, \boldsymbol{W}^\top \nabla L(\boldsymbol{W} \boldsymbol{H}) = \lambda \boldsymbol{H}. \tag{C.4}$$

If there does not exist $(\Delta \boldsymbol{W}, \Delta \boldsymbol{H})$ satisfying the requirement, we know that for any feasible curve $\phi(t) = (\boldsymbol{W}(t), \boldsymbol{H}(t))$ with $\phi(0) = (\boldsymbol{W}, \boldsymbol{H})$ on the sphere $\mathcal{S} = \{(\boldsymbol{W}', \boldsymbol{H}') : \|\boldsymbol{W}'\|_F^2 + \|\boldsymbol{H}'\|_F^2 = \|\boldsymbol{W}\|_F^2 + \|\boldsymbol{H}\|_F^2\}$, $t = 0$ admits the local minimum of $\mathcal{L}(\phi(t))$ and thus:

$$0 \leq \frac{d^2}{dt^2} \mathcal{L}(\phi(t))|_{t=0} = \phi'(0)^T \nabla^2 \mathcal{L}(\boldsymbol{W}, \boldsymbol{H}) \phi'(0) + \nabla \mathcal{L}(\boldsymbol{W}, \boldsymbol{H}) \phi''(0). \tag{C.5}$$

On the other hand, since the curve lies on the sphere $\mathcal{S}$, denote $h(\boldsymbol{W}, \boldsymbol{H}) = \|\boldsymbol{W}\|_F^2 + \|\boldsymbol{H}\|_F^2$, then $h(\phi(t))$ must stay as a constant, take twice derivative we have:

$$0 = \frac{d^2}{dt^2} h(\phi(t))|_{t=0} = \phi'(0)^T \nabla^2 h(\boldsymbol{W}, \boldsymbol{H}) \phi'(0) + \nabla h(\boldsymbol{W}, \boldsymbol{H}) \phi''(0). \tag{C.6}$$

Then sum these two conditions together, we have:

$$0 \leq \frac{d^2}{dt^2} (\mathcal{L}(\phi(t)) + \frac{|\lambda|}{2} h(\phi(t)))|_{t=0} = \phi'(0)^T \nabla^2 (\mathcal{L} + \frac{|\lambda|}{2} h)(\boldsymbol{W}, \boldsymbol{H}) \phi'(0) + \nabla (\mathcal{L} + \frac{|\lambda|}{2} h)(\boldsymbol{W}, \boldsymbol{H}) \phi''(0). \tag{C.7}$$

By equation (C.4) we know that $\nabla(\mathcal{L} + \frac{|\lambda|}{2} h)(\boldsymbol{W}, \boldsymbol{H}) = 0$ Note that $\phi'(0) \in \mathcal{T}(\boldsymbol{W}, \boldsymbol{H})$ since the curve lies on $\mathcal{S}$ and for any $(\Delta \boldsymbol{W}, \Delta \boldsymbol{H}) \in \mathcal{T}(\boldsymbol{W}, \boldsymbol{H})$ we can construct a curve $\phi(t)$ such that $\phi'(0) = (\Delta \boldsymbol{W}, \Delta \boldsymbol{H})$. Then (C.7) indicates that $\forall (\Delta \boldsymbol{W}, \Delta \boldsymbol{H}) \in \mathcal{T}(\boldsymbol{W}, \boldsymbol{H})$ we have:

$$0 \leq (\Delta \boldsymbol{W}, \Delta \boldsymbol{H})^\top \nabla^2 \mathcal{L}(\boldsymbol{W}, \boldsymbol{H})(\Delta \boldsymbol{W}, \Delta \boldsymbol{H}) + \frac{|\lambda|}{2} (\Delta \boldsymbol{W}, \Delta \boldsymbol{H})^\top \nabla^2 h(\boldsymbol{W}, \boldsymbol{H})(\Delta \boldsymbol{W}, \Delta \boldsymbol{H})$$
$$= (\Delta \boldsymbol{W}, \Delta \boldsymbol{H})^\top \nabla^2 \mathcal{L}(\boldsymbol{W}, \boldsymbol{H})(\Delta \boldsymbol{W}, \Delta \boldsymbol{H}) + |\lambda|(\|\Delta \boldsymbol{W}\|_F^2 + \|\Delta \boldsymbol{H}\|_F^2). \tag{C.8}$$

When $\lambda = 0$, by equation (C.3) we know that $\|\nabla L(WH)\|_2 > 0$, which gives us $|\lambda| \leq \|\nabla L(\boldsymbol{W} \boldsymbol{H})\|_2$ When $\lambda \neq 0$, combine the two equations in (C.4) we know:

$$\lambda \boldsymbol{W}^\top \boldsymbol{W} = \boldsymbol{W}^\top \nabla L(\boldsymbol{W} \boldsymbol{H}) \boldsymbol{H}^\top = \lambda \boldsymbol{H} \boldsymbol{H}^\top \Rightarrow \boldsymbol{W}^\top \boldsymbol{W} = \boldsymbol{H} \boldsymbol{H}^\top, \tag{C.9}$$

which further implies:
$$\|\boldsymbol{W}\|_F = \|\boldsymbol{H}\|_F, \quad \|\boldsymbol{W}\|_2 = \|\boldsymbol{H}\|_2.$$

Thus we also have (Note that when $\boldsymbol{W} = \boldsymbol{H} = 0$ we can take $\lambda$ to be zero):

$$\nabla L(\boldsymbol{W} \boldsymbol{H}) \boldsymbol{H}^\top = \lambda \boldsymbol{W} \Rightarrow |\lambda| \|\boldsymbol{W}\|_2 \leq \|\nabla L(\boldsymbol{W} \boldsymbol{H})\|_2 \|\boldsymbol{H}\|_2$$
$$\Rightarrow |\lambda| \leq \|\nabla L(\boldsymbol{W} \boldsymbol{H})\|_2.$$

Now when $|\lambda| < \|\nabla L(\boldsymbol{W} \boldsymbol{H})\|_2$, we can show that it will contradict with (C.8): We have shown that the rank of $\nabla L(\boldsymbol{Z})$ is $K - 1$, so by (C.4) and (C.9) there exist a vector $a$ such that $\boldsymbol{W} a = \boldsymbol{H}^\top a = 0$,

let $u$ and $v$ are the left and right singular vectors corresponding to the largest singular value of $\nabla L(\boldsymbol{W}\boldsymbol{H})$, construct $\Delta\boldsymbol{W} = ua^\top, \Delta\boldsymbol{H} = -av^\top$, then $(\Delta\boldsymbol{W}, \Delta\boldsymbol{H}) \in \mathcal{T}(\boldsymbol{W}, \boldsymbol{H})$ and:

$$(\Delta\boldsymbol{W}, \Delta\boldsymbol{H})^\top \nabla^2 \mathcal{L}(\boldsymbol{W}, \boldsymbol{H})(\Delta\boldsymbol{W}, \Delta\boldsymbol{H}) - \lambda(\|\Delta\boldsymbol{W}\|_F^2 + \|\Delta\boldsymbol{H}\|_F^2)$$
$$= (\boldsymbol{W}\Delta\boldsymbol{H} + \Delta\boldsymbol{W}\boldsymbol{H})\nabla^2 L(\boldsymbol{W}\boldsymbol{H})(\boldsymbol{W}\Delta\boldsymbol{H} + \Delta\boldsymbol{W}\boldsymbol{H}) + 2\langle\nabla L(\boldsymbol{W}\boldsymbol{H}), \Delta\boldsymbol{W}\Delta\boldsymbol{H}\rangle + |\lambda|(\|\Delta\boldsymbol{W}\|_F^2 + \|\Delta\boldsymbol{H}\|_F^2)$$
$$\leq 2\|a\|_2^2(|\lambda| - u^\top\nabla L(\boldsymbol{W}\boldsymbol{H})v) < 0.$$

Then it only remains to analyze the $|\lambda| = \|\nabla L(\boldsymbol{W}\boldsymbol{H})\|_2$ cases, construct another convex optimization problem:

$$\min_{\boldsymbol{Z}} L(\boldsymbol{Z}) + |\lambda|\|\boldsymbol{Z}\|_*, \tag{C.10}$$

suppose $\boldsymbol{Z}$ has SVD $\boldsymbol{Z} = \boldsymbol{U}\boldsymbol{\Sigma}\boldsymbol{V}^\top$, as we know that the subgradient of $\|\boldsymbol{Z}\|_*$ can be written as (see Watson (1992) for a proof):

$$\partial\|\boldsymbol{Z}\|_* = \left\{\boldsymbol{U}\boldsymbol{V}^\top + \boldsymbol{W}, \boldsymbol{W} \in \mathbb{R}^{K \times nK} \mid \boldsymbol{U}^\top\boldsymbol{W} = \boldsymbol{0}, \boldsymbol{W}\boldsymbol{V} = \boldsymbol{0}, \|\boldsymbol{W}\|_2 \leq 1\right\}. \tag{C.11}$$

On the other hand, we know that:

$$\boldsymbol{H}^\top\boldsymbol{H}\boldsymbol{H}^\top\boldsymbol{H} = \boldsymbol{H}^\top\boldsymbol{W}^\top\boldsymbol{W}\boldsymbol{H} = \boldsymbol{V}\boldsymbol{\Sigma^2}\boldsymbol{V}^\top$$
$$\boldsymbol{W}\boldsymbol{W}^\top\boldsymbol{W}\boldsymbol{W}^\top = \boldsymbol{W}\boldsymbol{H}\boldsymbol{H}^\top\boldsymbol{W}^\top = \boldsymbol{U}\boldsymbol{\Sigma^2}\boldsymbol{U}^\top,$$

which indicates that $\boldsymbol{H}^\top\boldsymbol{H} = \boldsymbol{V}\boldsymbol{\Sigma}\boldsymbol{V}^\top$ and $\boldsymbol{W}\boldsymbol{W}^\top = \boldsymbol{U}\boldsymbol{\Sigma}\boldsymbol{U}^\top$. Combine them with (C.4) we have:

$$\nabla L(\boldsymbol{W}\boldsymbol{H})\boldsymbol{H}^\top\boldsymbol{H} = \lambda\boldsymbol{W}\boldsymbol{H} \Leftrightarrow \nabla L(\boldsymbol{W}\boldsymbol{H})\boldsymbol{V}\boldsymbol{\Sigma}\boldsymbol{V}^\top = \lambda\boldsymbol{U}\boldsymbol{\Sigma}\boldsymbol{V}^\top$$
$$\Leftrightarrow \nabla L(\boldsymbol{W}\boldsymbol{H})\boldsymbol{V} = \lambda\boldsymbol{U}$$
$$\boldsymbol{W}\boldsymbol{W}^\top\nabla L(\boldsymbol{W}\boldsymbol{H}) = \lambda\boldsymbol{W}\boldsymbol{H} \Leftrightarrow \boldsymbol{U}\boldsymbol{\Sigma}\boldsymbol{U}^\top\nabla L(\boldsymbol{W}\boldsymbol{H}) = \lambda\boldsymbol{U}\boldsymbol{\Sigma}\boldsymbol{V}^\top$$
$$\Leftrightarrow \boldsymbol{U}^\top\nabla L(\boldsymbol{W}\boldsymbol{H}) = \lambda\boldsymbol{V}^\top.$$

Note that $|\lambda| = \|\nabla L(\boldsymbol{W}\boldsymbol{H})\|_2$, then by (C.11) we know that $-\nabla L(\boldsymbol{W}\boldsymbol{H}) \in |\lambda|\partial\|\boldsymbol{W}\boldsymbol{H}\|_*$. Then by the strict convexity of (C.10) we know $\boldsymbol{W}\boldsymbol{H}$ is the global minimum of it. In addition, we have $\|\boldsymbol{W}\|_F^2 + \|\boldsymbol{H}\|_F^2 = 2\mathrm{tr}\boldsymbol{\Sigma}^2) = 2\|\boldsymbol{W}\boldsymbol{H}\|_*$. In addition, previous works (Haeffele & Vidal, 2015) have shown that:

$$\|\boldsymbol{Z}\|_* = \min_{\boldsymbol{Z}=\boldsymbol{W}\boldsymbol{H}} \frac{1}{2}\left(\|\boldsymbol{W}\|_F^2 + \|\boldsymbol{H}\|_F^2\right),$$

which is equivalent to:

$$\|\boldsymbol{W}\boldsymbol{H}\|_* \leq \frac{1}{2}(\|\boldsymbol{W}\|_F^2 + \|\boldsymbol{H}\|_F^2).$$

Now for any $(\boldsymbol{W}', \boldsymbol{H}')$, we know that:

$$\mathcal{L}(\boldsymbol{W}, \boldsymbol{H}) + \frac{|\lambda|}{2}(\|\boldsymbol{W}\|_F^2 + \|\boldsymbol{H}\|_F^2) = L(\boldsymbol{W}\boldsymbol{H}) + |\lambda|\|\boldsymbol{W}\boldsymbol{H}\|_* \leq L(\boldsymbol{W}'\boldsymbol{H}') + |\lambda|\|\boldsymbol{W}'\boldsymbol{H}'\|_*$$
$$\leq \mathcal{L}(\boldsymbol{W}', \boldsymbol{H}') + \frac{|\lambda|}{2}(\|\boldsymbol{W}'\|_F^2 + \|\boldsymbol{H}'\|_F^2),$$

which indicates $(\boldsymbol{W}, \boldsymbol{H})$ must attain global minimum of the following optimization problem:

$$\min_{\boldsymbol{W}, \boldsymbol{H}} \mathcal{L}(\boldsymbol{W}, \boldsymbol{H}) + \frac{|\lambda|}{2}(\|\boldsymbol{W}\|_F^2 + \|\boldsymbol{H}\|_F^2).$$

If $(\boldsymbol{W}, \boldsymbol{H})$ does not satisfy neural collapse conditions, by the optimality of neural collapse solution (Theorem 3.3) we know there exists another point $(\boldsymbol{W}', \boldsymbol{H}')$ such that $\mathcal{L}(\boldsymbol{W}', \boldsymbol{H}') < \mathcal{L}(\boldsymbol{W}, \boldsymbol{H})$ and $\|\boldsymbol{W}\|_F^2 + \|\boldsymbol{H}\|_F^2 = \|\boldsymbol{W}'\|_F^2 + \|\boldsymbol{H}'\|_F^2$ thus:

$$\mathcal{L}(\boldsymbol{W}, \boldsymbol{H}) + \frac{|\lambda|}{2}(\|\boldsymbol{W}\|_F^2 + \|\boldsymbol{H}\|_F^2) > \mathcal{L}(\boldsymbol{W}', \boldsymbol{H}') + \frac{|\lambda|}{2}(\|\boldsymbol{W}'\|_F^2 + \|\boldsymbol{H}'\|_F^2),$$

which contradicts with the global optimality of $(\boldsymbol{W}, \boldsymbol{H})$, thus $(\boldsymbol{W}, \boldsymbol{H})$ must satisfy all of the neural collapse conditions and we finish the proof. □

# D ADDITIONAL EMPIRICAL RESULTS

**Gradient Descent on the ULPM Objective.** We conduct experiments on the ULPM objective (2.3) to support the results of convergence toward neural collapse in our theories. We set $N = 10$, $K = 5$, and $d = 20$, and used gradient descent with a learning rate of 5 to run $10^5$ epochs. We characterize the dynamics of the training procedure in Figure 2 based on four aspects: (1) Relative variation of the centered class-mean feature norms (i.e., $\text{Std}(\|\bar{\boldsymbol{h}}_k - \bar{\boldsymbol{h}}\|)/\text{Avg}(\|\bar{\boldsymbol{h}}_k - \bar{\boldsymbol{h}}\|)$) and the variation of the classifier's norms (i.e., $\text{Std}(\|\bar{\boldsymbol{w}}_k\|)/\text{Avg}(\|\bar{\boldsymbol{w}}_k\|)$). (2) Within-class variation of the last layer features (i.e., $\text{Avg}(\|\boldsymbol{h}_{k,i} - \boldsymbol{h}_k\|)/\text{Avg}(\|\boldsymbol{h}_{k,i} - \bar{\boldsymbol{h}}\|)$). (3) The cosines between pairs of last layer features (i.e., $\text{Avg}(|\cos(\bar{\boldsymbol{h}}_k, \bar{\boldsymbol{h}}_{k'}) + 1/(K-1)|)$) and that of the classifiers (i.e., $\text{Avg}(|\cos(\bar{\boldsymbol{w}}_k, \bar{\boldsymbol{w}}_{k'}) + 1/(K-1)|)$). (4) The distance between the normalized centered classifier and the normalized last layer feature (i.e., $\text{Avg}(|(\bar{\boldsymbol{h}}_k - \bar{\boldsymbol{h}})/\|\bar{\boldsymbol{h}}_k - \bar{\boldsymbol{h}}\| - \bar{\boldsymbol{w}}_k/\|\bar{\boldsymbol{w}}_k\||)$). Empirically, we observe that all four quantities decrease at approximately the rate $O(1/(\log(t)))$.

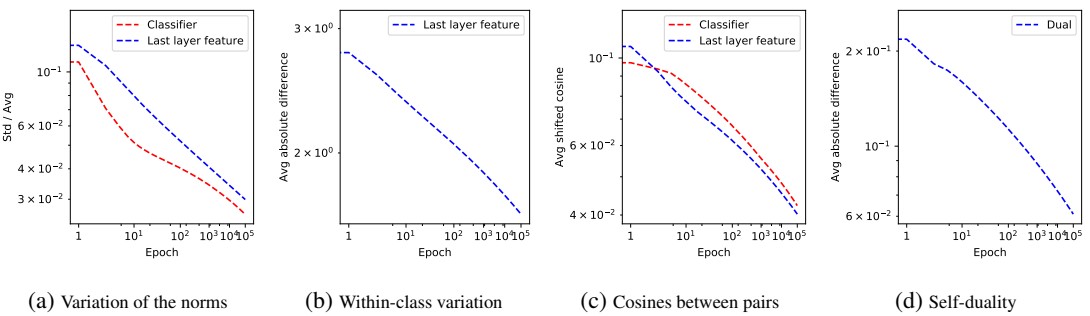

(a) Variation of the norms     (b) Within-class variation     (c) Cosines between pairs     (d) Self-duality

Figure 2: Training dynamics in ULPM. The $x$-axis in the figures is set to have $\log(\log(t))$ scales, and the $y$-axis in the figures are set to have $\log$ scales. (a) The dynamics of the variation of the centered class-mean features' norms (shown in blue) and the variation of the classifier's norms (shown in red). We observe that the logarithm of both terms decreases at a rate $O(1/(\log(t)))$. (b) Dynamics of within-class variation of the last layer features. The logarithm of the variation converges at approximately the rate $O(1/\log(t))$. (c) The dynamics of the cosines between pairs of last layer features (shown in blue) and those of the classifiers (shown in red). The logarithm of both terms converge approximately at rate $O(1/\log(t))$. (d) Dynamics of the distance between the normalized centered classifier and normalized last layer feature. The logarithm of the quantity converges at approximately the rate $O(1/\log(t))$ to the point of self-duality.

**Details of Realistic Training.** In the real data experiments, we trained the VGG-13 (Simonyan & Zisserman, 2014) and ResNet18 (He et al., 2016) on MNIST (LeCun et al., 1998), KMNIST (Clanuwat et al., 2018), FashionMNIST (Xiao et al., 2017) and CIFAR-10 datasets (Krizhevsky et al., 2009) without weight decay, and with a learning rate of 0.01, momentum of 0.3, and batch size of 128. The metrics are defined similarly to the ULPM case and the experiment results are reported in Figure 1, 3, 4, 5, 6, 7, and 8. All experiments were run in Python (version 3.6.9) on Google Colab. It

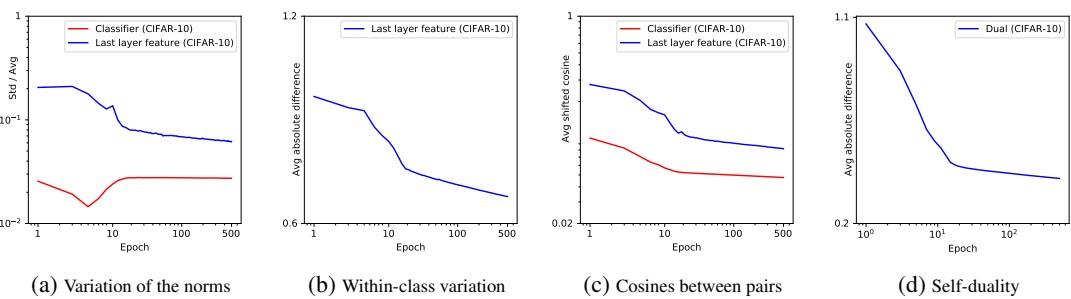

(a) Variation of the norms     (b) Within-class variation     (c) Cosines between pairs     (d) Self-duality

Figure 3: Experiments on real datasets without weight decay. We trained a VGG13 on CIFAR10 dataset. The $x$-axis in the figures are set to have $\log(\log(t))$ scales and the $y$-axis in the figures are set to have $\log$ scales.

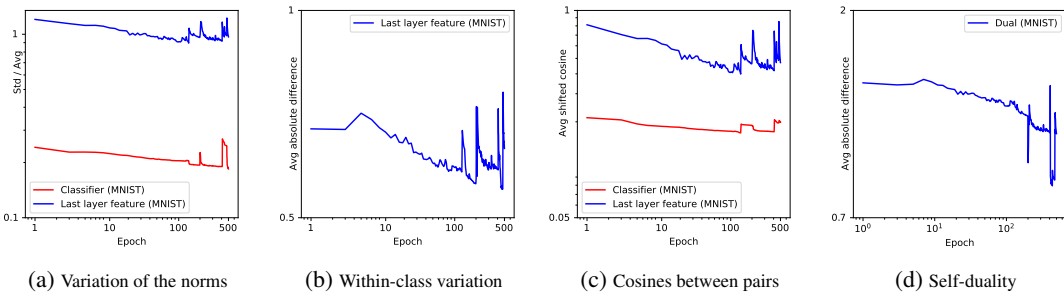

| | | | |
|---|---|---|---|
| (a) Variation of the norms | (b) Within-class variation | (c) Cosines between pairs | (d) Self-duality |

Figure 4: Experiments on real datasets without weight decay. We trained a VGG13 on the MNIST dataset. The $x$-axis in the figures are set to have $\log(\log(t))$ scales and the $y$-axis in the figures are set to have $\log$ scales.

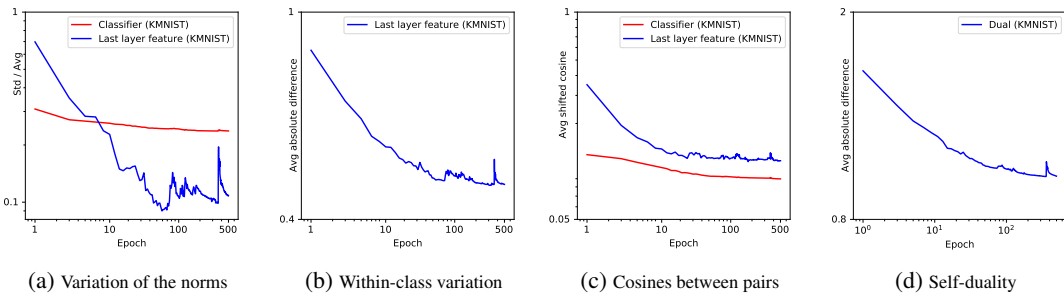

| | | | |
|---|---|---|---|
| (a) Variation of the norms | (b) Within-class variation | (c) Cosines between pairs | (d) Self-duality |

Figure 5: Experiments on real datasets without weight decay. We trained a VGG18 on the KMNIST dataset. The $x$-axis in the figures are set to have $\log(\log(t))$ scales and the $y$-axis in the figures are set to have $\log$ scales.

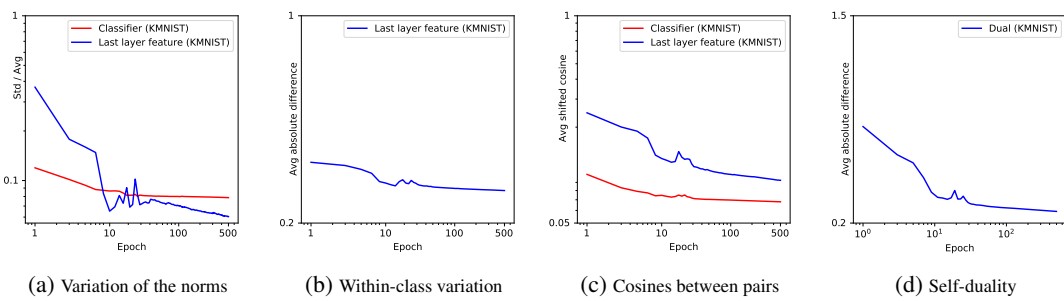

| | | | |
|---|---|---|---|
| (a) Variation of the norms | (b) Within-class variation | (c) Cosines between pairs | (d) Self-duality |

Figure 6: Experiments on real datasets without weight decay. We trained a ResNet18 on the KMNIST dataset. The $x$-axis in the figures are set to have $\log(\log(t))$ scales and the $y$-axis in the figures are set to have $\log$ scales.

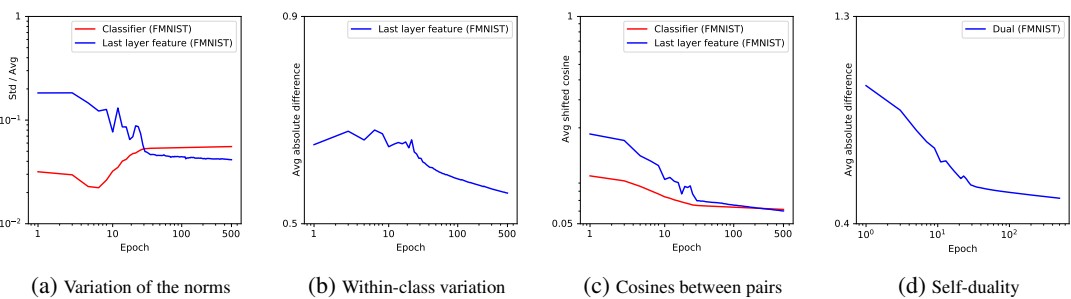

| | | | |
|---|---|---|---|
| (a) Variation of the norms | (b) Within-class variation | (c) Cosines between pairs | (d) Self-duality |

Figure 7: Experiments on real datasets without weight decay. We trained a VGG13 on the Fashion-MNIST dataset. The $x$-axis in the figures are set to have $\log(\log(t))$ scales and the $y$-axis in the figures are set to have $\log$ scales.

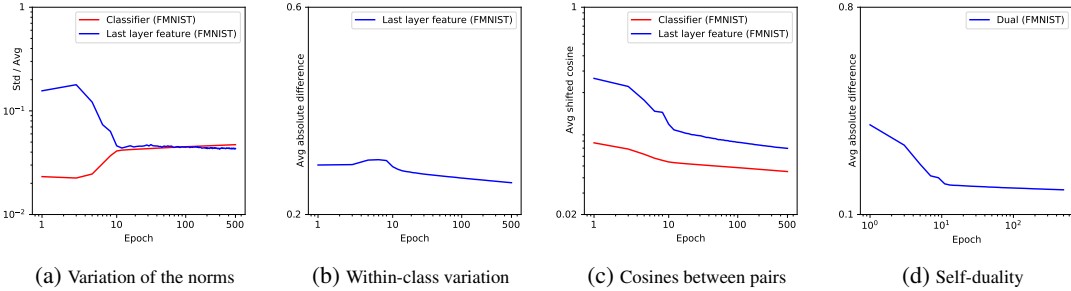

(a) Variation of the norms    (b) Within-class variation    (c) Cosines between pairs    (d) Self-duality

Figure 8: Experiments on real datasets without weight decay. We trained a ResNet18 on the Fashion-MNIST dataset. The $x$-axis in the figures are set to have $\log(\log(t))$ scales and the $y$-axis in the figures are set to have log scales.

should be mentioned that we can observe that the variation of classifier norm stays at a low value and do not decrease in some settings (e.g., Figure 7a and Figure 8a), this phenomenon is also found in Figure 2 of (Papyan et al., 2020), which might attribute to the network architecture (e.g., batch normalization) and characteristics of real-world datasets. Overall, we can find that neural collapse occurs for various network architectures and datasets under an unconstrained setting, which provides sound support for our theory.

## E  LIMITATION AND FUTURE DIRECTIONS

Currently, our analysis is still limited in the terminal phase of training and only studies the training behavior after data is perfectly separated. Such assumption is necessary for all implicit regularization analysis (Lyu & Li, 2019; Ji et al., 2020; Nacson et al., 2019a) under a nonlinear setting, and how to fully characterize the early time training dynamics is still an open problem. Moreover, we have provided a loss landscape analysis in the paper showing that there is no spurious minimum in the tangent space. To guarantee the training dynamics do not get stuck in saddle points and converge to neural collapse solution, we need to introduce randomness in the training dynamics (e.g. use stochastic gradient descent (Ge et al., 2015)), otherwise simple gradient flow may get stuck in saddle points as we have shown in Example 3.1. How to characterize the training dynamics of stochastic gradient descent in the nonlinear setting is also an open problem. It is an interesting direction to further build a complete characterization of convergence to neural collapse solutions by addressing the early time training dynamics and studying the stochastic gradient descent.

