# OpenReview forum: "An Unconstrained Layer-Peeled Perspective on Neural Collapse"
_ICLR.cc/2022/Conference — ICLR 2022 Poster_

### Official Review · Reviewer_ez35 · 2021-10-19

**Correctness:** 4
**Technical Novelty And Significance:** 3
**Empirical Novelty And Significance:** 3
**Recommendation:** 8
**Confidence:** 3

**Main Review:**

Overall, this is a good paper and the result is interesting. The paper first shows a connection between the limiting point of gradient flow on the cross-entropy loss and the KKT solution of a minimum-norm separation problem. It also shows that every global optimal point of the minimum-norm separation problem satisfies the neural collapse condition. To deal with the concern that a KKT solution is not necessarily a global optimal point of the minimum-norm separation problem, a second-order landscape analysis of the minimum-norm problem is provided to show that at all the other KKT points of the minimum-norm separation problem, there is a direction that leads to a lower objective (of the Lagrangian of the minimum-norm problem).

I think a clarification is needed on whether the paper shows that gradient flow on ULPM is guaranteed to return a solution that is a global optimum point of the minimum-norm separation problem, i.e. is guaranteed to converging to a point that exhibits the neural collapse.

Typo: On the second line of the proof of Theorem 3.1, there is a typo regarding the definition of $\tilde{w}_i$: $\frac{1}{n} \rightarrow \frac{1}{ K}$.


**Summary Of The Paper:**

This paper shows that for the cross-entropy loss, when updating both the weight vector and the features by gradient flow, the final solution exhibits the neural collapse behavior, where the neural collapse refers to an observation pointed out by (Papyan 2020), which is that (NC1) the within-class features collapse to their class mean, (NC2) the average of the features of each class converges to having the equal length and all pairs of the class-means have equal-size angles, (NC3) the weight and the class-mean are aligned, (NC4) the weight converges to the nearest class-center of the features.
The optimization problems (or the simplified model) in  the previous works that theoretically show the neural collapse require some constraints or regularization, which are seldom used in practice. The optimization model (ULPM) in this paper does not require these, and so is much closer to the realistic (of training deep networks).


**Summary Of The Review:**

(see above)

---

> ### Author Response · Authors · 2021-11-22
> **Response to Reviewer ez35**
>
> We are really grateful for your comments and suggestions. Currently, the result of our paper is still limited in the terminal phase of training and only studies the training behavior after data is perfectly separated. Such assumption is necessary for all implicit regularization analysis [1,2,3] under a nonlinear setting, and how to fully characterize the early time training dynamics is still an open problem. Moreover, we have provided a loss landscape analysis in the paper showing that there is no spurious minimum in the tangent space. To guarantee the training dynamics do not get stuck in saddle points and converge to neural collapse solution, we need to introduce randomness in the training dynamics (e.g. use stochastic gradient descent [4]), otherwise simple gradient flow may get stuck in saddle points as we have shown in Example 3.1. How to characterize the training dynamics of stochastic gradient descent in the nonlinear setting is also an open problem. We believe that our paper has provided insightful views and make significant contributions on how gradient descent is biased toward neural collapse solution, and it is an interesting direction to further build a complete characterization of convergence to neural collapse solutions by addressing the early time training dynamics and studying the stochastic gradient descent. In the updated version, we have added this discussion in Section E of the appendix.
>
> [1]Kaifeng Lyu and Jian Li. Gradient descent maximizes the margin of homogeneous neural networks. arXiv preprint arXiv:1906.05890, 2019.
>
> [2]Ji, Ziwei, and Matus Telgarsky. "Directional convergence and alignment in deep learning." arXiv preprint arXiv:2006.06657 (2020).
>
> [3]Nacson, Mor Shpigel, et al. "Lexicographic and depth-sensitive margins in homogeneous and non-homogeneous deep models." International Conference on Machine Learning. PMLR, 2019.
>
> [4]Ge R, Huang F, Jin C, et al. Escaping from saddle points—online stochastic gradient for tensor decomposition[C]//Conference on learning theory. PMLR, 2015: 797-842.

---

### Official Review · Reviewer_Au7e · 2021-11-02

**Correctness:** 3
**Technical Novelty And Significance:** 3
**Empirical Novelty And Significance:** 2
**Recommendation:** 8
**Confidence:** 4

**Main Review:**

This is an interesting contribution. The problem analyzed in this paper is important, and their technique based on the optimization landscape coupled with the connection between the min-norm classifier and neural collapse is insightful. In addition, the paper is well written and clearly organized.

My main concerns are the following:

- The narrative of the paper seems to be directed towards conveying the message that their studied model (without norm constraints) is more realistic, and in several places they comment on how other models studied in related works are "further from practical networks". I find these comments at least exaggerated, but potentially also inaccurate: in the studied model, the norms of the classifier and features diverge (they only converge in direction), which does not happen in practical deep learning models displaying neural collapse. As a result, one could also argue that the presented model is further from practical settings.. To be clear: I'm not trying to argue this is the case. I'm simply suggesting that trying to make a point along these lines sounds counter-productive, and the contribution of this paper would be better served without these subjective distractions.

- I am slightly confused about the claim of the authors that they present a global analysis of the optimization landscape: it seems like their result holds for a proper initialization that has a loss function less than log(2), which implies that the features and classifiers are already separable. I agree with the authors that this is still relevant and interesting, as it precisely captures the behavior observed in the final stages of training. However, if this is the case, then the term "global analysis" is inaccurate, and this holds only upon a correct initialization. Could the authors explain why they refer to this analysis as global?

- The empirical results on Fig.1 are slightly confusing to me: there are two different colors dedicated to "last layer features", in different graphs. Why is this so? Also, as far as I can tell, the variance of the class-specific classifiers does not seem to converge in either case, which contradicts the presented results. It is true that "they stay within small values", but this is different from the convergence results that the authors present as their main result. Can the authors explain this?

Minor comments:
- Page 2: "Finally, We verify.. "
- page 7: "unoptimal points" -> non-optimal
- Theorem 3.4 "is (or is not) neural collapse solution" -> "is (or is not) a neural collapse solution".








**Summary Of The Paper:**

This paper analyzes the phenomenon of neural collapse from the simplified perspective of an unconstrained features models, whereby the only two optimization variables in the model are the last layer classifier and features, which are fit to some labels by minimizing the cross entropy loss. The authors leverage their observation that maximizing the margin results in a model achieving neural collapse, and then study the corresponding problem of minimum-norm classifier, and show that gradient flow converges directionally to the neural collapse solution. The authors then show that the optimization landscape is benign, in that all critical points that are not global minimum have a decreasing direction in the tangent space.

**Summary Of The Review:**

I believe this is a valuable contribution, but have some reservations about the presentation of the results. I would be happy to increase my rating after reading the responses by the authors.

---

> ### Author Response · Authors · 2021-11-22
> **Response to Reviewer Au7e**
>
> We are really grateful for your comments and suggestions. In the following, we address your concerns point by point.
>
> - Regarding the concerns about our narrative, we think our formulation is closer to practice for the following reasons:
>
>   1. Feature norm regularization or constraint is still not equivalent to weight decay on parameters in deep neural networks.
>   2. In modern practice, the weight decay is usually chosen to be very small (e.g. 1e-4 or 5e-4), which leads to a weak regularization effect, and the training dynamics may not be affected much to be biased toward neural collapse. Thus we aim to study the unconstrained setting to show that the emergence of neural collapse should be attributed to properties of cross-entropy and gradient descent rather than explicit regularization.
>   3. Recent empirical observations [1] showed that neural networks continue to perform well without weight decay. Moreover, in [2], it's proven that training with an exponential learning rate schedule is equivalent to training with weight decay. As a result, we believe that our unconstrained setting can be a good approximation to real-world deep neural networks.
>
>   After receiving kind suggestions from reviewers, we also realize that some of our statements might not be precise. **We have made some revisions to the new version.**
>
> - Regarding the concerns about the global landscape analysis, the reviewer may get confused about assumptions across different theorems. In this paper we do provide a **global analysis** of the optimization landscape, please notice that we **only** require the log(2) condition in training dynamics analysis (Theorem 3.2) and we **do not** assume it in landscape analysis (Theorem 3.4). This condition is necessary for all training dynamics analysis on nonlinear models [3,4,5], and how to study the early time training dynamics is still an open problem.
>
> - Regarding the concerns about the empirical result, we are sorry for using different colors, it actually refers to the same criterion in different figures. For the non-convergence behavior of classifiers,  the non-decreasing variation of norms is also found in [6] (see Figure 2 in [6] for details, you can find a non-decreasing behavior of norm variation under various settings), which may attribute to the network architecture like batch-normalization and characteristics of real-world datasets. It is a very interesting direction to further explore why this phenomenon occurs in real-world deep learning. We have added some **additional experiment results** in both Section 4 and Section D in the appendix, where we can find a convergence behavior for variation of classifier norm under many settings. Moreover, we would like to refer to Figure 2 in Section D, where we conduct numerical experiments on the ULPM objective and the results are consistent with our theory.
>
> We hope that our response has addressed your concerns. We would very much appreciate it if you could reconsider your score based on the response. Please let us know if you have any further questions and we are happy to clarify. Thank you!
>
> [1] Chiyuan Zhang, Samy Bengio, Moritz Hardt, Benjamin Recht, and Oriol Vinyals. Understanding deep learning requires rethinking generalization. arXiv preprint arXiv:1611.03530, 2016.
>
> [2]Li Z, Arora S. An exponential learning rate schedule for deep learning[J]. arXiv preprint arXiv:1910.07454, 2019.
>
> [3]Ji, Ziwei, and Matus Telgarsky. "Directional convergence and alignment in deep learning." arXiv preprint arXiv:2006.06657 (2020).
>
> [4]Nacson, Mor Shpigel, et al. "Lexicographic and depth-sensitive margins in homogeneous and non-homogeneous deep models." International Conference on Machine Learning. PMLR, 2019.
>
> [5]Ge R, Huang F, Jin C, et al. Escaping from saddle points—online stochastic gradient for tensor decomposition[C]//Conference on learning theory. PMLR, 2015: 797-842.
>
> [6]Vardan Papyan, XY Han, and David L Donoho. Prevalence of neural collapse during the terminal phase of deep learning training. Proceedings of the National Academy of Sciences, 117(40):24652–24663, 2020.

---

### Official Review · Reviewer_sXBT · 2021-11-03

**Correctness:** 4
**Technical Novelty And Significance:** 3
**Empirical Novelty And Significance:** 2
**Recommendation:** 6
**Confidence:** 4

**Main Review:**

## Strengths
This paper studies neural collapse, a widely observed phenomenon in training deep neural networks. Existing work on the analysis of neural collapse is based on either constraints or regularizers on the features and classifiers. The authors analyze the gradient flow of the ULMP without any constraints or regularizers, and provide second-order analysis of the objective function, ensuring gradient descent converges to neural collapse solutions.

## Weaknesses
1. Weight decay is often employed in training neural networks. This paper studies the training problem without any regularizers, but it is unclear why this formulation is closer to the practice than the one with regularizers.

2. Eq. (3.3) only shows that a non-optimal solution is not a local minimum. It may be a higher-order saddle. If the results show the (Riemannian) Hessian has a negative eigenvalue, then I think this is a much stronger result and should be described in Theorem 3.4.

2. As mentioned in the paper, the training dynamics will diverge to infinity where the cross-entropy loss archives its minimum. Thus, is possible that eq. (3.3) holds for any finite $(W,H)$ (if we allow any direction for $(\Delta W,\Delta H)$), i.e., any finite $(W,H)$ is not a local solution?

3. Which manifold does the tangent space in Definition 3.1 refers to? In eq. (3.3), why highlight the direction on the tangent space?
In general, the tangent space is used in the manifold optimization where the variables are constrained on the manifolds, but here the training loss has no constraints. Can you clarify this? Also in Remark 3.4, it says Hessian matrix. Is it Riemannian Hessian?

**Summary Of The Paper:**

This paper provides new analyses for understanding the neural collapse phenomenon observed in training deep learning classifiers. In particular, to understand the last-layer features and classifiers, the authors study the gradient flow of the unconstrained layer-peeled model (ULPM) without any regularizers and show that gradient flow converges to neural collapse classifiers and features. Experiments are provided to demonstrate the neural collapse phenomenon when no weight decay is used.

**Summary Of The Review:**

Overall, this paper provides new analyses for understanding the neural collapse phenomenon. The analyzed formulation removes the constraints and regularizers on the features and constraints, which is different from the existing approach.

---

> ### Author Response · Authors · 2021-11-22
> **Response to Reviewer sXBT**
>
> We are really grateful for your comments and suggestions. In the following, we address your concerns point by point.
>
> - Regarding the concerns about why our formulation is more realistic, we think our formulation is closer to practice for the following reasons:
>
>   1. Feature norm regularization or constraint is still not equivalent to weight decay on parameters in deep neural networks.
>   2. In modern practice, the weight decay is usually chosen to be very small (e.g. 1e-4 or 5e-4), which leads to a weak regularization effect, and the training dynamics may not be affected much to be biased toward neural collapse. Thus we aim to study the unconstrained setting to show that the emergence of neural collapse should be attributed to properties of cross-entropy and gradient descent rather than explicit regularization.
>   3. Recent empirical observations [1] showed that neural networks continue to perform well without weight decay. Moreover, in [2], it's proven that training with an exponential learning rate schedule is equivalent to training with weight decay. As a result, we believe that our unconstrained setting can be a good approximation to real-world deep neural networks.
>
>   After receiving kind suggestions from reviewers, we also realize that some of our statements might not be precise. **We have made some revisions to the new version.**
>
> - Regarding the concerns about the higher-order saddle, in this paper, we have proven that the Hessian has a negative eigenvalue in the first equation of page 25, which further shows that the non-optimal solution can not be a higher-order saddle. We are grateful for the reviewer's suggestion and have added a remark to make this fact more clear in the new version.
>
> - Regarding the concerns about local solutions, as stated in the second paragraph of Section 3.2, we study the directional convergence in the unconstrained setting. As a result, we do not consider growing along the radial direction and study the optimality of different directions in the tangent space. Such directional convergence has been widely studied in implicit regularization literature [3,4,5], and we advanced their theory by analyzing the landscape and showing that there is no spurious minimum in the tangent space. Our results close the gap between KKT points and global optimum and give an explicit characterization of global optimum via neural collapse.
>
> - Regarding the concerns about tangent space, since the ULPM objective will always decrease along the direction of the current point and the optimum is attained only in infinity, we study the optimality of direction and do not consider growing along the radial direction. In this case, we consider the optimality on a sphere. By Taylor’s expansion, we only need to analyze the gradient and Hessian in such tangent space to obtain the optimality on the sphere. This fact has been included and proven rigorously in the appendix and we would like to refer to lines between equation (C.4) and (C.8) in the appendix for a formal discussion. The idea is that the sphere is a smooth manifold that can be locally approximated by the tangent space at each point. In Remark 3.4, it should be Riemannian Hessian rather than standard Hessian since we ignore the radial direction, we apologize for this mistake and will fix it in the new version.
>
> We hope that our response has addressed your concerns. We would very much appreciate it if you could reconsider your score based on the response. Please let us know if you have any further questions and we are happy to clarify. Thank you!
>
> [1] Chiyuan Zhang, Samy Bengio, Moritz Hardt, Benjamin Recht, and Oriol Vinyals. Understanding deep learning requires rethinking generalization. arXiv preprint arXiv:1611.03530, 2016.
>
> [2]Li Z, Arora S. An exponential learning rate schedule for deep learning[J]. arXiv preprint arXiv:1910.07454, 2019.
>
> [3]Kaifeng Lyu and Jian Li. Gradient descent maximizes the margin of homogeneous neural networks. arXiv preprint arXiv:1906.05890, 2019.
>
> [4]Ji, Ziwei, and Matus Telgarsky. "Directional convergence and alignment in deep learning." arXiv preprint arXiv:2006.06657 (2020).
>
> [5]Nacson, Mor Shpigel, et al. "Lexicographic and depth-sensitive margins in homogeneous and non-homogeneous deep models." International Conference on Machine Learning. PMLR, 2019.

---

> > ### Comment · Reviewer_sXBT · 2021-11-29
> > **Thank you for the response**
> >
> > I appreciate the great efforts the authors have made in response to my comments. My concerns have been well addressed. I still think that the tangent space in Definition 3.1 comes out of the blue since the original problem has no constraints. It will be helpful to incorporate the above discussions to motivate the description of tangent space, Riemannian Hessian, and related concepts into the final version.

---

### Official Review · Reviewer_PTW6 · 2021-11-03

**Correctness:** 3
**Technical Novelty And Significance:** 3
**Empirical Novelty And Significance:** 2
**Recommendation:** 6
**Confidence:** 3

**Main Review:**

Strengths:
- The paper writing is good, and the presentation is clear
- The theoretical part is interesting, which builds connection between max-margin implicit bias and NC, and the theories are seriously developed

Weaknesses:
- I find the empirical result section, i.e. Section 4, is quite weak. It only contains a Figure 1, which itself is not well-presented since it is hard for me to gain much insights. This section really needs to be improved.
- Some statements are debatable. In the first bullet point of the main contributions, it says "feature regularization is unrealistic and never used in practice", but the L-2 norm regularization, or weight decay, is widely used in practice. In fact, in the experiments of [Papyan et al. 2020], it uses such weight decay by setting the parameter to be 1e-4 or 5e-4. Therefore, in Table 1, I don't think the "feature norm regularization" spot for [Papyan et al. 2020] should have a cross mark there.
- The "Notations" section in the bottom of page 4 can be moved earlier; otherwise many notations are used widely before without introduction, e.g. [K]


Typos:
- In the second line after Eq. (1.1), the "y_i" is unbolded while I think it should be bolded?
- In the paragraph above Eq. (2.3), the notation "N" should be "n"? since we are using "n" in Section 2.1
- In the first equation in Section 3.1, "k \in [1,K], i \in [1,n]" --> "k \in [K], i \in [n]"


**Summary Of The Paper:**

The paper studies the recently discovered Neural Collapse (NC) phenomenon for deep neural network training using Cross Entropy (CE) loss. It theoretically analyzes the problem with a so-called unconstrained layer-peeled model (ULPM), and then it shows that the ULPM with CE loss has a benign landscape.

**Summary Of The Review:**

Overall, despite its weakness as I described above, this paper does provide some interesting approach in understanding the NC with CE loss. Considering its theoretical nature and the serious study, I would say it is marginally above the acceptance threshold.

---

> ### Author Response · Authors · 2021-11-22
> **Response to Reviewer PTW6**
>
> We are really grateful for your comments and suggestions. In the following, we address your concerns point by point.
>
> - Regarding the concerns about the empirical result section, we are sorry for any confusion caused. In the new version, we have polished the presentation and added some additional experiments and explanations. We hope they can help the reviewer understand our results better and gain more insights from them.
>
> - Regarding the concerns about our statements, we say "feature regularization is unrealistic and never used in practice" for the following reasons:
>
>   1. Feature norm regularization or constraint is still not equivalent to weight decay on parameters in deep neural networks.
>   2. In modern practice, the weight decay is usually chosen to be very small (e.g. 1e-4 or 5e-4), which leads to a weak regularization effect, and the training dynamics may not be affected much to be biased toward neural collapse. Thus we aim to study the unconstrained setting to show that the emergence of neural collapse should be attributed to properties of cross-entropy and gradient descent rather than explicit regularization.
>   3. Recent empirical observations [1] showed that neural networks continue to perform well without weight decay. Moreover, in [2], it's proven that training with an exponential learning rate schedule is equivalent to training with weight decay. As a result, we believe that our unconstrained setting can be a good approximation to real-world deep neural networks.
>
>   Also, we mark a cross on "feature norm regularization" for [3] since weight decay is not equivalent to feature norm regularization. After receiving kind suggestions from reviewers, we also realize that some of our statements might not be precise. **We have made some revisions to the new version.**
>
> - Regarding the concerns about the notation, we thank the reviewer for the kind suggestions, and we will move the notation part to the beginning of Section 2 to make it more clear.
>
> - Regarding the minor typos, we are grateful for the careful correctness. We will fix these minor typos and apologize for any confusion it caused.
>
> We hope that our response has addressed your concerns. We would very much appreciate it if you could reconsider your score based on the response. Please let us know if you have any further questions and we are happy to clarify. Thank you!
>
> [1] Chiyuan Zhang, Samy Bengio, Moritz Hardt, Benjamin Recht, and Oriol Vinyals. Understanding deep learning requires rethinking generalization. arXiv preprint arXiv:1611.03530, 2016.
>
> [2]Li Z, Arora S. An exponential learning rate schedule for deep learning[J]. arXiv preprint arXiv:1910.07454, 2019.
>
> [3]Vardan Papyan, XY Han, and David L Donoho. Prevalence of neural collapse during the terminal phase of deep learning training. Proceedings of the National Academy of Sciences, 117(40):24652–24663, 2020.

---

### Author Response · Authors · 2021-11-22
**An updated version of the paper**

We thank all the reviewers for their helpful feedback. We have uploaded a revised paper based on their suggestions, and the major revisions are summarized below:

- We reorganized the empirical results section to make it more clear, and we added more experiment results in Section D.
- We changed some inaccurate statements about our unconstrained settings compared with feature regularization.
- We added some discussion about the limitation of our analysis and future directions in Section E.
- We fixed some minor typos in this paper.

---

### Decision · Program_Chairs · 2022-01-20

**Decision:**

Accept (Poster)

**Comment:**

This paper follows the recent line of work of theoretically analyzing the Neural Collapse phenomenon, by making certain simplifying assumptions on the problem setup. In this case, the authors use cross-entropy loss on an unconstrained model where second-to-last representations become free variables. Their main results characterise the NC as the only global minimiser.
Reviewers were positive about this work, and concluded it presents a valuable addition to the growing analysis of NC. They also pointed out several clarity issues that should be addressed in the final revision, including a more objective comparison to prior work. Ultimately this work will be an interesting addition to the conference, and therefore the AC recommends acceptance.